# A downward counterfactual analysis of flash floods in Germany

Paul Voit[1] and Maik Heistermann[1]

[1]Institute for Environmental Sciences and Geography, University of Potsdam, Potsdam, Germany

**Correspondence:** Paul Voit (voit@uni-potsdam.de)

**Abstract.** Counterfactuals are scenarios that describe alternative ways of how an event, in this case an extreme rainfall event, could have unfolded. In this study, we present the results of a counterfactual search for flash flood events in Germany. We used a radar-based precipitation dataset of the German weather service to identify the ten most extreme precipitation events in Germany from 2001 to 2022, and then assumed that any of these top 10 events could have happened anywhere in Germany. In other words, the events were shifted around all over Germany. For all resulting positions of the precipitation fields, we simulated the corresponding peak discharge for any affected catchment smaller than 750 km$^2$. From all the realisations of this simulation experiment, the maximum peak discharge was identified for each catchment.

In a case study, we first focused on the devastating flood event in July 2021 in western Germany. We found that a moderate shifting of the event in space could change the event peak flow at the gauge Altenahr by a factor of two. Compared to the peak flow of 1004 m$^3$/s caused by the event in its original position, the worst case counterfactual of that event led to a peak flow of 1311 m$^3$/s. Shifting another event that had occurred just one month earlier in eastern Germany over the Ahr river valley even effectuated a simulated peak flow of 1651 m$^3$/s.

For all analysed subbasins in Germany, we found that, on average, the highest counterfactual peak exceeded the maximum original peak (between 2001 and 2022) by a factor of 5.3. For 98 % of the basins, the factor was higher than 2.

We discuss various limitations of our analysis, which are important to be aware of: with regard to the quantification and selection of candidate rainfall events, the hydrological model, and the design of the counterfactual search experiment. Still, we think that these results might help to expand the view on what could happen in case certain extreme events occurred elsewhere, and thereby reduce the element of surprise in disaster risk management.

## 1 Introduction

Flash floods constitute a relevant natural hazard in many regions of the world. In comparison to river floods, the footprint of a flash flood event is small, yet the local impact can be devastating. Flash floods combine low predictability, erratic overflow behavior, high flow velocities and often massive debris loads. They are mainly caused by heavy precipitation events (HPEs) with very high rainfall intensities, and characterized by a rapid concentration of runoff. Usually, flash floods are defined by a response time of less then six hours (Borga et al., 2008; Marchi et al., 2010) which mostly confines their occurrence to catchments smaller than 1000 km$^2$. The underlying HPEs often are highly variable in space and time (Borga et al., 2008). In addition to the properties of the HPE itself, the geographical context governs the nature of the hydrological response and thus

the resulting impact. Hence, both atmospheric and hydrological processes interact across various spatial and temporal scales during flash floods (Georgakakos, 1986).

The management of flash flood risks often requires corresponding extreme value statistics. The robustness of such statistics is contingent upon the length of historical records (Woo, 2019), and might be compromised by the effects of ongoing climate change. Locally, flash floods are rare events; observational data is scarce as the affected catchments are typically small and ungauged (Gaume et al., 2008). This makes it difficult to establish reliable extreme value statistics for many locations. Worst case flood scenarios and their dependence on spatio-temporal characteristics of precipitation as well as the catchment's hydrological conditions have not yet been fully understood (Zischg et al., 2018; Marchi et al., 2010). Spatio-temporal patterns of rainfall and their dynamic interaction with topography and land use significantly influence the generation and propagation of flood peaks (Beven and Hornberger, 1982; Singh, 1997; Tarolli et al., 2013; Emmanuel et al., 2015; Zischg et al., 2018). This implies that even slight changes in event realizations could significantly affect the response. Yet, the sample size of investigated HPEs is often limited.

To enhance our understanding of the flash flood hazard in Germany, we adopt an approach known as "counterfactual thinking" (Roese, 1997; Woo, 2019) which was also proposed recently by Montanari et al. (2023) in the context of flood research. This approach involves considering alternative ways of how events *could* have unfolded. For risk assessment, downward counterfactuals are particularly interesting: they involve thought experiments about past events with worse outcomes than what actually transpired (Roese, 1997). Such thought experiments can provide valuable insights into worst-case scenarios that have not (yet) occurred. This way, the level of preparedness could be increased, although the approach typically cannot underpin such worst-case scenarios with occurrence probabilities.

Spatial changes, in particular, play a significant role in counterfactual analysis (Woo, 2019): the coincidence of an HPE with an area characterized by steep slopes, impervious surfaces, and multiple stream intersections can trigger very high flood peaks, which would be less pronounced in less steep and more natural catchments.

Based on 16 years of radar observations, Lengfeld et al. (2019) found that extreme daily precipitation is dependent on the orography but that heavy hourly rainfall can occur anywhere in Germany. Based on the – admittedly strong – assumption that historical HPEs could have happened anywhere in Germany, we propose, in this study, a systematic downward counterfactual search for flash floods in Germany. To that end, we adopted the following approach:

1. Based on radar-based precipitation estimates from 2001–2022, we created a catalog of HPEs in Germany and ranked these HPEs by using a recently proposed metric to assess the extremity of rainfall across spatial and temporal scales (Voit and Heistermann, 2022).

2. We shifted the 10 most extreme HPEs from our catalog to each subbasin in Germany and simulated the corresponding quick runoff (QR) response for the whole affected area. This way we created a total of 23,000 counterfactual scenarios for each HPE. Each of these scenarios includes the QR simulations for hundreds of subbasins.

3. Additionally we model, for each subbasin, the QR response to *all* events contained in our catalog, in their original position. The corresponding results serve as a reference for the maximum historical QR response in each subbasin, to which we compare the results of the counterfactual search.

Based on this groundwork, we first investigate, in a regional case study, counterfactual scenarios of the devastating July 2021 precipitation event over the Ahr river catchment (see Mohr et al., 2023, for details). We then expand our analysis to all of Germany, and explore the potential hydrological response to rare HPEs in case they had happened anywhere in Germany, and search for downward counterfactual scenarios. Based on this search, we try to answer how close actual historical events (within the last 22 years) have already touched upon the worst case scenario, and discuss the usefulness of this information for flood risk management.

## 2 Data

In this section, we will describe the data that was used for the extraction of HPEs as well as the data sources for our hydrological model. The overall study area is Germany. We will also present a case study in which we focus on the catchment of the Ahr river down to the runoff gauge at Altenahr. In our hydrological model, this catchment consists of 37 subbasins (details of this case study are presented in section 4.2). Both the overall study area as well as the case study area are illustrated in Fig. 1.

### 2.1 Precipitation Data

To allow for a detailed representation of the spatio-temporal variability of rainfall, we used the radar climatology product (RADKLIM v2017.002) provided by Germany's national meteorological service (Deutscher Wetterdienst; DWD hereafter) between 2001 and 2022. RADKLIM is a reprocessed version of the operational radar-based quantitative precipitation estimation (QPE) product (RADOLAN, see Winterrath et al., 2012) of the DWD since 2001. To minimize the occurrence of artifacts (Lengfeld et al., 2019) and to allow for heavy rainfall analysis (Kreklow et al., 2019), the radar data is adjusted by additional rainfall data from gauges (hourly and daily), a homogeneous set of algorithms and advanced climatological corrections (Winterrath et al., 2018a). RADKLIM represents the Germany-wide hourly precipitation at a resolution of 1 x 1 km . Some parts of Germany (very North, South and East) have not been covered by radar since 2001, but overall data coverage over Germany is good with less than 10 % missing hours in most areas (Lengfeld et al., 2019). The RADKLIM data set is available on the DWD open data server (Winterrath et al., 2018b). We would like to emphasize the importance of using radar-based precipitation products when dealing with flash floods: Compared to rain gauge interpolation, the error of radar-based products have been shown to be considerably smaller (Journée et al., 2023). Zoccatelli et al. (2010) also showed that the errors of rain gauge interpolations for flash-flood triggering HPEs do not average out at the spatial scales associated to flash floods.

### 2.2 Digital elevation model

The Digital Elevation Model over Europe (EU-DEM) was used to delineate catchments in Germany and for further analysis of runoff concentration (flow paths and traveltime to catchment outlets). For the EU-DEM, SRTM (Shuttle Radar Topography

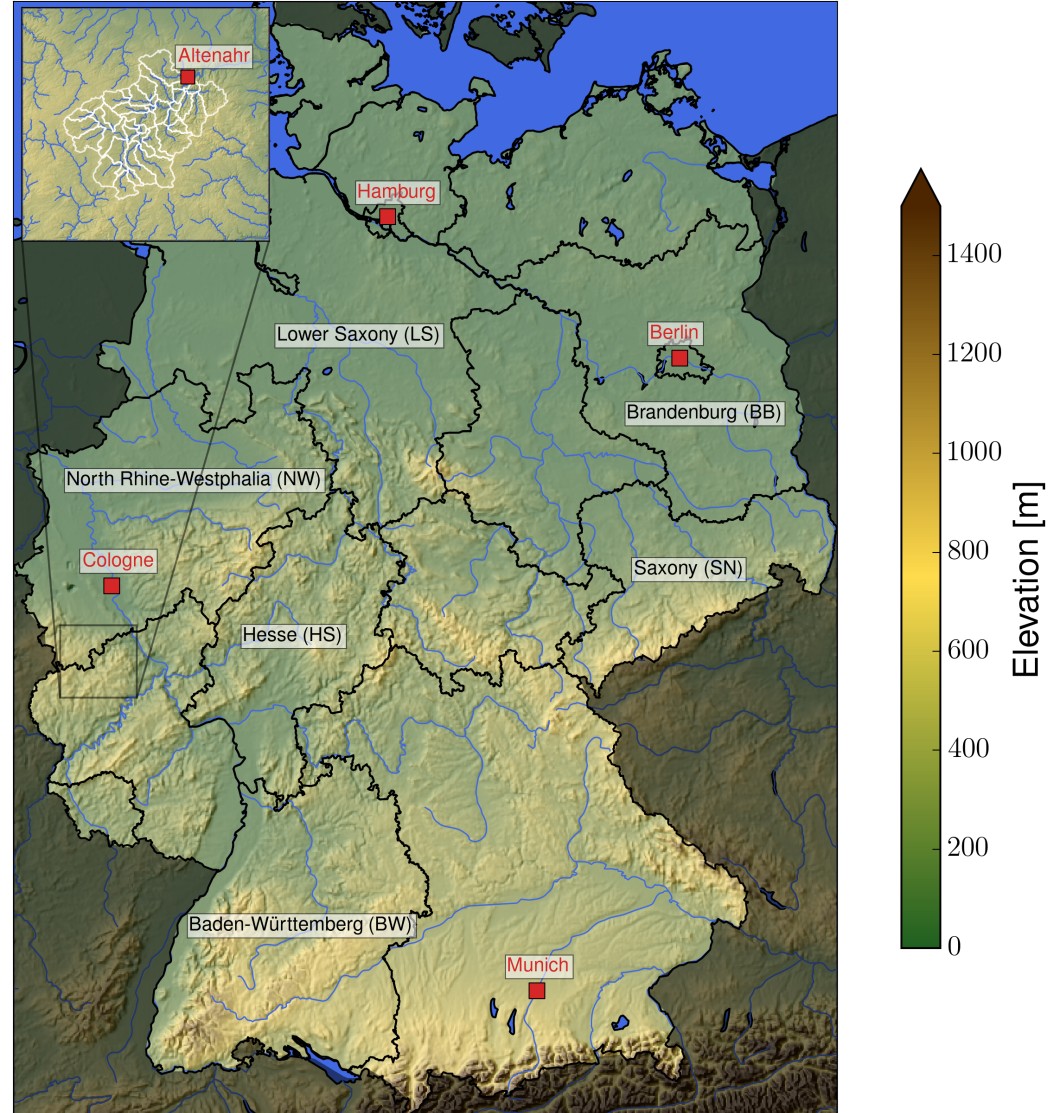

**Figure 1.** Map of the study region (Germany): topography, major water bodies (blue), federal states (black), and selected cities (red); white: subbasins of the Ahr catchment upstream of Altenahr (case study region, see section 4.2).

Mission) and ASTER GDEM (Advanced Spaceborne Thermal Emission and Reflection Radiometer Global Digital Elevation Model) data is fused by a weighted averaging approach . The data set has a spatial resolution of 25 m and can be downloaded from the Copernicus Land Monitoring service (European Commission, 2016).

## 2.3 Land cover

Information about land cover was derived from CORINE CLC5-2018 (BKG, 2018). The product is based on a classification of high resolution satellite data into 37 land cover classes (for Germany), according to the nomenclature of the European Environmental Agency (EEA). Objects with a minimum size of 5 ha are considered in the classification and the product is updated every three years.

## 2.4 Soil data

Soil information was derived from the "BUEK 200" (national soil survey at a scale of 1:200,000; BGR, 2018) which is compiled from the surveys of each federal state at a scale of 1:200,000 by the Federal Institute for Geosciences and Natural Resources (Bundesanstalt für Geowissenchaften und Rohstoffe, BGR) in cooperation with the National Geological Services (Staatliche Geologische Dienste, SGD). For each mapping unit, the BUEK200 provides areal fractions of dominant soil types and the corresponding profile information, including texture, bulk density and many more.

## 3 Methods

This section describes the methods used to create a catalog of HPEs, an outline of the hydrological model to model the formation and concentration of quick runoff, and the design of the counterfactual simulation experiment.

## 3.1 Catalog of heavy rainfall events in Germany

While the DWD provides a catalog of HPEs (CatRaRE: Catalog of Radar-based Heavy Rainfall Events; Lengfeld et al., 2021), we still opted to develop our own catalog. This decision was motivated by the fact that HPEs which exhibit extreme behaviour across various durations and spatial scales can trigger different flood mechanisms that can intersect and amplify each other. For instance, high-intensity rainfall on a small spatial scale may be embedded within larger events and preceded by periods of low-intensity rainfall that increase soil moisture. Antecedent soil moisture has a significant impact on event runoff coefficients and is essential for flash flood modelling (Marchi et al., 2010). To that end, Voit and Heistermann (2022) have recently proposed a new metric, the cross-scale weather extremity index (xWEI), to detect and assess HPEs that were extreme at various spatial and temporal scales. Both the WEI (as used by the CatRaRE catalog) and the xWEI quantify a measure of extremeness along two dimensions: rainfall duration and spatial extent. Hence the variation of extremeness along these dimensions could be illustrated as a surface. While the WEI corresponds to the maximum value of that surface, the xWEI corresponds to the volume under the surface, meaning that it is high if the extremeness is high across spatial and temporal scales.

The catalog was created by applying a multi-step procedure. Considering the RADKLIM dataset as a 3-D array (one temporal dimension, two spatial dimensions), we first apply a moving 3-D window (72 hours x 3 km x 3 km) to the entire dataset. Within this moving window, the rainfall extremeness is computed for each voxel and for various durations. Afterwards, a clustering

algorithm is applied to identify spatio-temporal clusters of extreme rainfall. The details of this approach together with an illustration are provided in the supplementary material Appendix A. The resulting catalog contains 17302 events.

## 3.2 Modelling quick runoff

We used standard GIS techniques (sink filling, flow accumulation, flow direction and catchment delineation) implemented in the Python package PCRaster (Karssenberg et al., 2010) to derive the subbasins. Since our model requires the areal average precipitation per subbasin as input, the subbasins need to be sufficiently small to represent the effects of spatial rainfall variability on the formation and concentration of quick runoff. For that purpose, we set outlet points for the subbasins at every stream intersection with a Strahler order of 7 or larger. This way we divided the study area into 22384 subbasins. For the analysis we

restricted our modelling to a spatial scale of up to 750 $km^2$ which leads to 19809 remaining basins. The median basin size is 12 $km^2$ (25th percentile: 6.9 $km^2$, 75th percentile: 20.2 $km^2$). Figure B1 (supplementary) illustrates the distribution of subbasin sizes as a histogram.

In the case study (section 4.2) we focused on the catchment of Altenahr (Rhineland-Palatinate) as a study region (see Fig. 1). The city of Altenahr was heavily affected by the so-called "Bernd"-event in July 2021 in western Germany and hit by a flood

on 15th July 2021 that caused massive destruction. The catchment upstream of Altenahr, before the inflow of the Vischelbach, has an approximate size of 728.6 $km^2$ and is, in our model, split into 37 subbasins. The smallest subbasin has a size of 3 $km^2$, the largest 48 $km^2$ and the median size is 17.1 $km^2$. The average curve number for the whole catchment is 66 (see section 3.2.1), varying between 61-72 for the individual subbasins (all values for medium soil moisture, soil moisture class 2).

Flash floods are characterized by quick (surface or near-surface) runoff components (Georgakakos, 1986; Marchi et al.,

2010; Grimaldi et al., 2010; Borga et al., 2014). Thus, the hydrological model setup can be simplified, as processes like evaporation and groundwater dynamics have minimal impact on the peak formation. While the formation of quick runoff is mostly controlled by soil conditions and land-use, the concentration of quick runoff is primarily driven by topographic relief (Ruiz-Villanueva et al., 2012). Based on these considerations, we adopt the following hypotheses for our model:

- Flash floods peaks are dominated by quick runoff (Marchi et al., 2010; Borga et al., 2014).

- The morphology and topography of the catchment exerts the main control on the concentration of quick runoff.

- Flash floods occur predominantly in small to medium-sized catchments with an area smaller than 750 km².

- Evapotranspiration and baseflow dynamics are negligible.

- The objective of the model is not to accurately simulate discharge dynamics. Instead, our focus is primarily on the timing and magnitude of the quick runoff peak flow (QR) and making relative comparisons between different counterfactuals

and original events.

- Due to the lack of accurate streamflow data (Gaume et al., 2004; Borga et al., 2014) and the computational effort to model a large number of counterfactual scenarios, we cannot use a model that requires parameter calibration.

To this end, our model consists of only two components which are described in more detail in the following subsections below:

1.  The Curve Number (CN) method (U.S. Department of Agriculture-Soil Conservation Service, 1972; Natural Resources Conservation Service, 2004; Garen and Moore, 2005), calculates the effective rainfall based on land use, soil characteristics and antecedent rainfall.

2.  The geomorphological instanteous unit hydrograph (GIUH) method represents the concentration of quick runoff for each subbasin. By superimposing these hydrographs, we can efficiently analyze a large number of counterfactual precipitation
scenarios.

With increasing catchment size, the influence of channel mechanics and hydro-engineering on stream flow becomes more important. Due to the limitations of our model, we are unable to incorporate these factors. Consequently, we restrict our QR modeling to subbasins with a spatial scale of up to 750 $km^2$. The majority of the 19809 remaining subbasins are head catchments (13741) and have an average size of 15 $km^2$ and a median size of 11.2 $km^2$.

### 3.2.1  SCS-CN method

We use the established SCS-CN (curve number) method (U.S. Department of Agriculture-Soil Conservation Service, 1972; Ponce and Hawkins, 1996; Natural Resources Conservation Service, 2004) to calculate the effective precipitation depending on soil, land use and antecedent wetness. For each subbasin, we utilized the "BUEK 200" soil database (see section 2.4) to obtain the fractions of four different soil classes (from permeable to non-permeable). This classification was combined with
the CORINE CLC5-2018 land use data (see section 2.3). Given that flash flood events primarily occur during summer months (see section 3.3), we made slight adjustments to the CN values for agricultural areas to account for the influence of summer crops (based on Seibert et al., 2020). Ultimately, a single CN value was calculated for each subbasin using a weighted areal average.

Rainfall series for each individual subbasin and event realization was obtained using the zonal statistics functionality of the
Python package "wradlib" (Heistermann et al., 2013) which computes the weighted average rainfall per subbasin based on the intersection of each RADKLIM pixel with the subbasin. This areal-averaged rainfall data was then used to calculate the effective rainfall using the SCS-CN method.

### 3.2.2  GIUH

To route the effective rainfall derived from the SCS-CN method to the subbasin outlet, we utilized the GIUH-method. Especially
for ungauged basins, this method provides a simple and widely used tool for rainfall-runoff modeling by taking into account the geomorphological features of a basin (Singh et al., 2014; Yi et al., 2022). The GIUH method constructs a hydrograph by estimating the travel time of an instantaneously applied unit of effective rainfall (typically 1 mm) from each grid cell in the catchment to the outlet.

The travel time is determined based on the length of surface flow paths to the outlet and the corresponding flow velocities. Various methods exist to calculate flow velocities. We opted for the spatially distributed travel time model introduced by Maidment et al. (1996) which allows for the use of distributed terrain information in an efficient manner (Bunster et al., 2019). This model demonstrated suitability in a comparative study conducted by Grimaldi et al. (2010). In this method, the flow velocity in a cell is defined as a function of the contributing upstream area $A$ and the local slope $s$:

$$v = v_m \frac{s^b A^c}{[s^b A^c]_m} \tag{1}$$

with $v$ as the velocity assigned to a cell with local slope $s$ and upstream drainage area is $A$. For $b$ and $c$, 0.5 has been proven to be a suitable value (Maidment et al., 1996; Grimaldi et al., 2010). $v_m$ describes the average value of the velocity in all cells in the watershed and is set to 0.1 m/s based on the study of Grimaldi et al. (2010). $[s^b A^c]_m$ is the watersheds average value of the slope-area term. By incorporating the drainage area $A$ into the formula, this method considers the increasing hydraulic radius (Manning's equation) with higher flow volume, thereby capturing the downstream increase in flow velocity without the need to estimate roughness coefficients for individual grid cells. Furthermore, it eliminates the need to differentiate between hill slope and channel grid cells within the catchment. Similarly to previous studies (Sivapalan et al., 2002; Marchi et al., 2010; Creutin et al., 2013), we constrained the resulting velocities within the range of 0.06 m/s to 3 m/s. By summing the velocities of each grid cell along a flow path, we estimated the travel time for each cell to reach the outlet using the *ldddist* function from the Python PCRaster package (Karssenberg et al., 2010). The hydrograph, representing the QR response of the catchment, is then derived by the probability density function of travel times from all grid cells to the catchment outlet. This method assumes a time- and discharge-invariant velocity field, allowing for a convolution of the GIUHs to model the catchment response to the effective rainfall of an HPE.

In the case that two subcatchments flow together we add the hydrograph (superposition) of the upstream basin to the hydrograph of the downstream basin with a temporal delay. The delay is determined by the travel time from the inlet of the downstream basins to its outlet.

### 3.3 Design of the downward counterfactual simulation experiment

For our counterfactual study, we selected the ten highest-ranking events from our catalog (Tab. 1). We then relocated each of these events to each subbasin in Germany. Since the spatial extent of the events is much larger than that of the subbasins, we aligned the pixel with the highest hourly rainfall with the centroid of the corresponding subbasin. We then modelled the QR response for all sub-basins within the HPE's bounding box (not just for the subbasin to which we shifted the centroid of the HPE). That way, the overall results are not too sensitive to how we actually align an HPE with an individual subbasin. By following this procedure, we generated approximately 230,000 counterfactual QR scenarios across Germany (23,000 sub-basins multiplied by 10 HPEs with their centroids shifted across all sub-basins). These data sets contain a total of more than 829 million counterfactual QR hydrographs for the individual subbasins and we refer to them as "cf_germany". Additionally,

we filtered the complete "cf_germany" dataset by limiting the maximum distances over which the HPEs were shifted to 10, 20, 50 and 250 km. We refer to these filtered datasets as cf_10km, cf_20km, cf_50km, and cf_250km.

## 3.4 Metrics for flash flood response

To compare flood peaks across different basin sizes, we utilized the concept of the unit peak discharge (UPD) (refer to Castellarin (2007) for a summary of the concept). The UPD $[\mathrm{m}^3/\mathrm{s}/(\mathrm{km}^2)^{0.6}]$ is the ratio between the discharge peak$[\mathrm{m}^3/\mathrm{s}]$ and the
reduced upstream catchment area $[(\mathrm{km}^2)^{0.6}]$. To limit the influence of the upstream catchment area, we use an exponent of 0.6 (similarly to Gaume et al., 2008; Emmanuel et al., 2017). Amponsah et al. (2018) used a UPD of 0.5 $\mathrm{m}^3/\mathrm{s}/\mathrm{km}^2$ (which corresponds to 0.66 $\mathrm{m}^3/\mathrm{s}/(\mathrm{km}^2)^{0.6}$) as the lower threshold for the definition of flash floods across a variety of climates and studies in their flash flood catalog. To illustrate the unit of the UPD: a UPD of 3 $\mathrm{m}^3/\mathrm{s}/(\mathrm{km}^2)^{0.6}$ could equal an 18 $\mathrm{m}^3/\mathrm{s}$ flood peak in a basin of 20 $\mathrm{km}^2$ size, or a peak flow of 72 $\mathrm{m}^3/\mathrm{s}$ in a 200 $\mathrm{km}^2$ basin.

## 4   Results and discussion

In this section, we present the results of our analysis. Section 4.1 starts by introducing the ten most severe precipitation events which were identified based on the cross-scale extremity index. By shifting them all over Germany, they form the basis of our spatial counterfactual search experiment. The hydrological simulation results of this experiment are first explored in a case study for the Ahr catchment, and put into context to the devastating flood event in July 2021 4.2. Second, we summarize the
results of our simulation experiment for all of Germany.

## 4.1   Top 10 HPEs

In this section, we introduce the ten most severe precipitation events between 2001 and 2022, based on DWD's RADKLIM dataset. These events are the basis of our counterfactual simulation experiment.

The ten most extreme events in our HPE catalog all occurred during the summer months, and are displayed in Figure 2 and
Table 1.

It should be noted that the xWEI is sensitive to the spatial extent of an event. Therefore, the top 10 events are generally very large. The catalog might contain events that are more severe at small spatio-temporal scales, say at the scale of small headwater catchments. The resulting limitations for our analysis will be further discussed in Sect. 5.1. However, events with a large spatial extent and a large xWEI value are likely to include smaller event clusters that are extreme at smaller spatio-temporal scales
which exactly motivated the choice to rank events by the xWEI (see also Sect. 3.1). Nonetheless, future applications might choose different catalogs or different metrics and ranking criteria to select candidate events for a counterfactual search.

Very different levels of impacts were reported for these events. In section C of the supplementary material, we put each event in context to other available references (scientific or media), and also attempt to compile estimates of reported damages and loss of lives, if available. While all ten events featured exceptional amounts of rainfall and a corresponding runoff response, only five
of them caused massive impacts (SN/Aug02, SN/Jun13, BW/May16, BB/Jun17, and, with by far the highest impact, NW/Jul21)

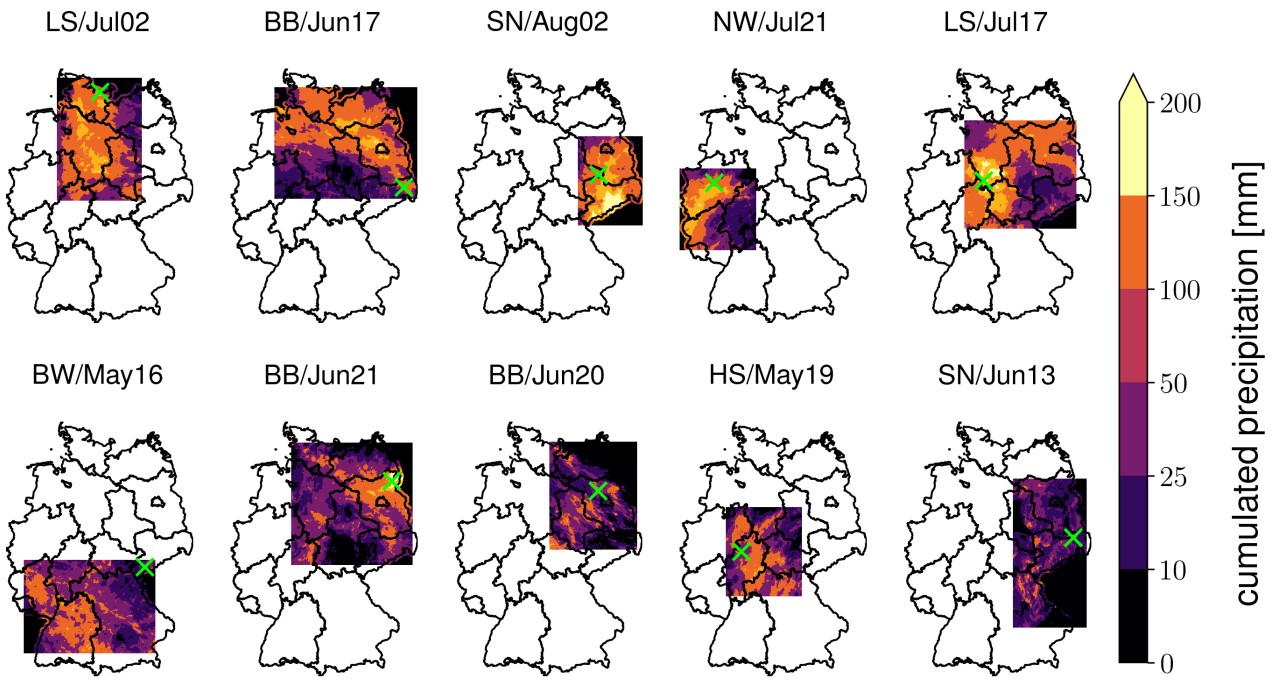

**Figure 2.** Original position and cumulated precipitation of the 10 most extreme HPEs from the event catalog. The green cross indicates the location of the highest hourly precipitation during the event which we chose as centroid when shifting the events to create counterfactuals.

**Table 1.** The ten most extreme HPEs from our catalog. The ID was constructed from an acronym that specifies the federal state in which the event mainly occurred, the month, and the year (starting from the year 2000). The precipitation values in the table [mm] are based on a 10 x 10 km moving window average, the ranking is based on the xWEI metric.

| rank | ID | Date | xWEI | max. 1h prec. | max. 24h prec. | max. 72h prec. | location of max. 1h prec. |
|------|-----|------|------|---------------|----------------|----------------|---------------------------|
| 1 | LS/Jul02 | Jul 15-20, 2002 | 4148 | 41 | 138 | 154 | Plön |
| 2 | BB/Jun17 | Jun 26-Jul 2, 2017 | 3901 | 44 | 149 | 157 | Bautzen |
| 3 | SN/Aug02 | Aug 10-15, 2002 | 3741 | 23 | 224 | 255 | Wittenberg |
| 4 | NW/Jul21 | Jul 11-16, 2021 | 3542 | 35 | 136 | 150 | Dortmund |
| 5 | LS/Jul17 | Jul 22-27, 2017 | 3327 | 24 | 136 | 233 | Göttingen |
| 6 | BW/May16 | May 27-Jun 1, 2016 | 3304 | 53 | 98 | 106 | Erzgebirgskreis |
| 7 | BB/Jun21 | Jun 28-Jul 3, 2021 | 3235 | 38 | 217 | 219 | Uckermark |
| 8 | BB/Jun20 | Jun 11-16, 2020 | 2964 | 53 | 93 | 104 | Ostprignitz-Ruppin |
| 9 | HS/May19 | May 18-23, 2019 | 2718 | 26 | 106 | 114 | Waldeck-Frankenberg |
| 10 | SN/Jun13 | Jun 18-22, 2013 | 2575 | 62 | 121 | 121 | Bautzen |

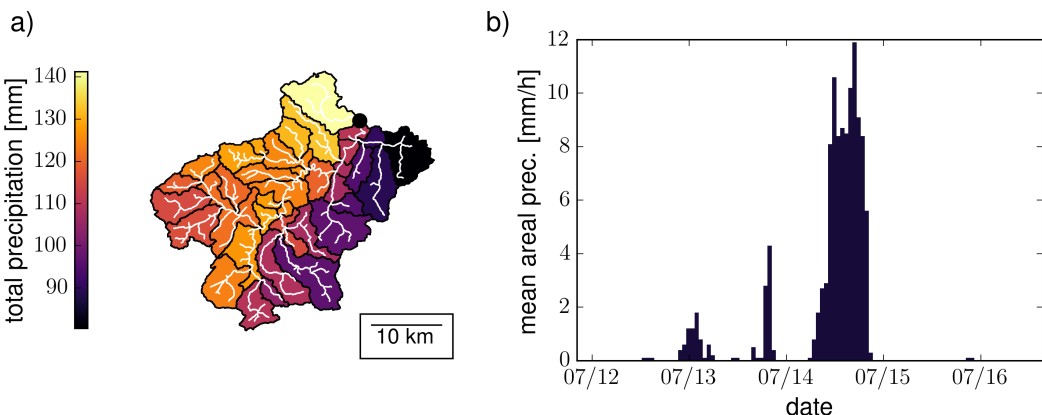

**Figure 3.** Total rainfall estimates (RADKLIM) for the original NW/Jul21 event for the Altenahr catchment: a) total rainfall [mm] in the Altenahr subbasins, b) areal average of precipitation [mm/h] for the Altenahr catchment. The outlet of the catchment, is shown in black, subbasin borders in black, streams in white.

while for the remaining events (LS/Jul02, LS/Jul17, HS/May19, BB/Jun20 and BB/Jun21), the impact was apparently not high enough to attract attention beyond the affected regions. The results of the counterfactual scenario analysis, as presented in the following, should help to understand whether the different levels of impacts for these events were mainly caused by their specific geographic position.

## 4.2 Case study: Altenahr

Before exploring the results for all of Germany, we zoom into the counterfactual scenarios obtained for the Ahr catchment (Fig. 3a). The Ahr was the most severely affected river during the July 2021 floods over western Germany (see Mohr et al., 2023, for more background on the flood event and the Ahr catchment). Typically, a flash flood is characterized by a lag time of up to six hours between the centroid of the effective rainfall and the hydrograph peak (Borga et al., 2008; Marchi et al., 2010; Morin et al., 2002). So, strictly speaking, the flood event at Altenahr does not qualify as a flash flood: according to our model, the lag time at Altenahr amounted to approximately eight hours. Still, the event at Altenahr is a highly illustrative example for a swift and massive runoff response at the meso-scale which is the result of the temporal superposition of various upstream flash floods. In fact, all 23 sub-basins upstream of Altenahr show a lag time of less than 6 hours, 22 of them even less than 3 hours.

By shifting around the top 10 HPEs (as listed in Tab. 1) over Germany, we created a total of 38,871 counterfactual rainfall scenarios over the Altenahr catchment, representing a large variety of spatial rainfall patterns and average rainfall totals, for all of which we simulated the QR peak flow. In the following, we compare these counterfactual peak flows to the peak simulated for the NW/Jul21 event in its original position. Any event label *NW/Jul21_x* will refer to a spatial counterfactual of the NW/Jul21 event. The same naming convention is adopted for the other events from Table 2.

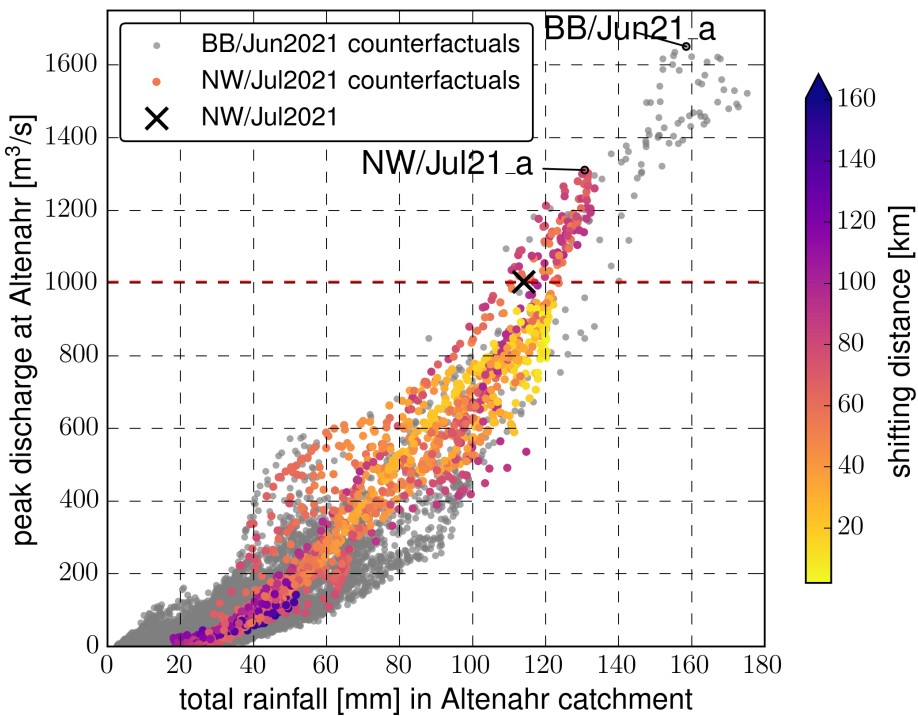

**Figure 4.** Total rainfall amount and resulting QR peak for counterfactuals of the NW/Jul21 (yellow to blue) and BB/Jun2021 (grey) HPEs for the Altenahr catchment. Black cross: areal mean of total rainfall the catchment received during the event and the resulting runoff for the event in its original spatial position. The point color of the NW/Jul21 counterfactuals indicates the distance to the centroid of the original NW/Jul21 event.

**Table 2.** Selected counterfactuals for the Altenahr catchment.

| ID | QR peak [m$^3$/s] | total prec. [mm] | lat. centroid | lon. centroid |
|---|---|---|---|---|
| NW/Jul21 | 1004 | 114 | 50.740 | 6.965 |
| NW/Jul21_a | 1311 | 131 | 51.315 | 7.519 |
| BB/Jun21_a | 1651 | 159 | 50.437 | 6.792 |

Figure 4 illustrates the results from the counterfactual study for the Altenahr catchment. The total rainfall for the catchment
for each counterfactual and the resulting highest QR peak is shown. Despite the positive correlation ($r^2 = 0.96$, Fig.4) between total rainfall and resulting flood peaks we notice that the same total rainfall amounts can yield markedly diverse QR peaks.

During the original event (NW/Jul21), the Altenahr catchment received an areal rainfall average of approximately 114 mm of which 98 mm fell within 12 hours on the 14th of July. The maximum hourly areal average was 12 mm (Fig. 3b). This

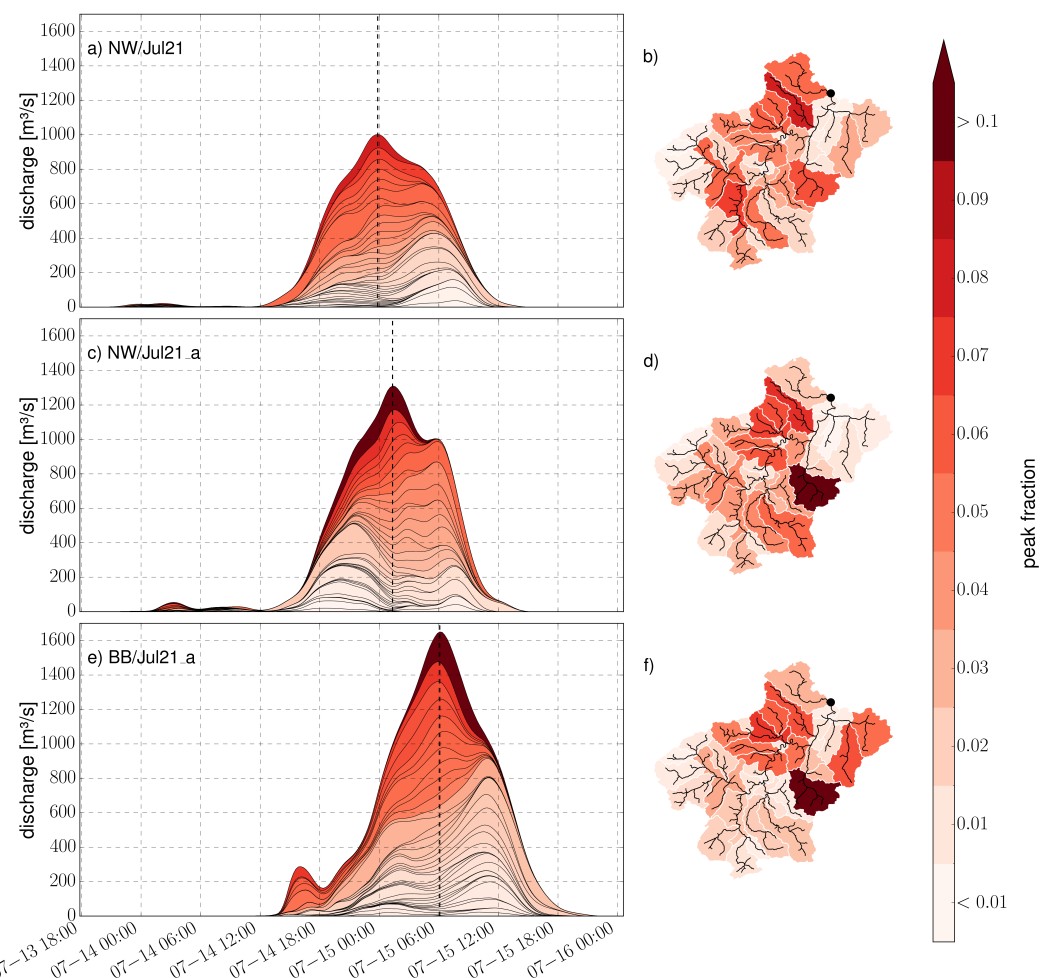

**Figure 5.** Contributions of individual subbasins to the runoff peak at Altenahr for three scenarios. The left side shows the superposition of runoff from the subbasins. The color code describes the runoff contribution to the peak flow (white = low, red = high, dotted line = peak position). On the right side, the same color code is used to display the spatial distribution of the contributions of each subbasins. Streams are shown in black as well as the outlet at Altenahr (black dot). Each row of the plot shows a different precipitation scenario a, b) original NW/Jul21 event, c, d) NW/Jul21_a counterfactual, e, f) BB/Jun21_a counterfactual (see also Tab. 2).

rainfall results in a modelled QR peak of 1004 $\mathrm{m}^3/\mathrm{s}$. Our model experiment illustrates that, for this specific amount of total areal rainfall ($114 \pm 1$ mm), the QR peaks span a range of 536 to 1090 $\mathrm{m}^3/\mathrm{s}$ across all NW/Jul21 counterfactuals (Fig. 4). This signifies that, with an identical total rainfall volume, the QR peak can vary by a factor of 2.

The original event's QR peak is already substantial; however, 6 % of the NW/Jul21 counterfactuals would have caused an even higher QR peak. All of these downward counterfactuals were created by a spatial shift of the original event by 45 - 97 km. The maximum modelled QR is 1311 $\mathrm{m}^3/\mathrm{s}$ (NW/Jul21_a), which is considerably higher than the 1004 $\mathrm{m}^3/\mathrm{s}$ peak resulting

from the original event. This outcome would have been achieved if the centroid of NW/Jul21 would have been shifted by only 75 km.

Fig. 5a) and b) illustrates, for the original NW/Jul21 event, the superposition of peaks at the gauge Altenahr from the discharge of the individual subbasins. The maximum counterfactual rainfall total (130.7 mm for NW/Jul21_a) results in a modelled QR peak of 1311 $m^3/s$ (Fig. 5c and d). Altogether, these cases underpin the importance of the spatio-temporal event structure for the peak discharge formation. The mean total precipitation for the whole Altenahr catchment conceals the spatio-temporal distribution of rainfall among its subbasins. In our model, the catchment consist of 37 subbasins (Fig. 3a).

By spatially shifting the other nine HPEs from Table 1 across Germany, we can get an idea of the kind of QR flood peaks that these HPEs could have triggered at Altenahr - had they happened in the region. The BB/Jun21 event is an interesting case: this event happened just one month prior to NW/Jul21 in the north-east of Germany (Uckermark). Although rated almost as extreme as the NW/Jul21 event (Tab. 1), it caused little damage in its original position. However, various spatial positions of this event would have apparently caused even higher QR peaks in Altenahr, up to 1651 $m^3/s$ (BB/Jun21_a, Fig. 5e and f). Among all ten events, the BB/Jun21 counterfactuals lead to the highest modelled QR peaks for the gauge Altenahr.

Out of all counterfactuals, 1 % resulted in QR peaks higher than the one from the original event NW/Jul21. This underlines the rarity of the event. Among these, there are no counterfactuals of the events BW/May16, BB/Jun17, LS/Jul17, HS/May19 and BB/Jun20. Further investigation is needed to understand the differences in the spatio-temporal structure of these events and how these HPEs were different to the other top 10 events to understand why these HPEs did not have the potential to create any maximum counterfactual peaks.

In summary, the analysis of 38,871 QR counterfactuals for the Altenahr catchment has demonstrated that, while the original NW/Jul21 event was exceptional, numerous spatial constellations of the same event and especially of the BB/Jun21 event could have led to higher flood peaks. While the areal average rainfall total is a key control on peak formation, the spatio-temporal distribution of this total can moderate flood peak formation substantially.

The discharge and timing of the modeled QR peak for the NW/Jul21 event (1004 $m^3/s$) fits well with recent reconstructions that estimated a peak flow around 1000 $m^3/s$ at Altenahr (Mohr et al., 2023). This is surprising given that the RADKLIM product might underestimate the event rainfall (Saadi et al., 2023). In any case, our model confirms that the NW/Jul21 event triggered a swiftly moving flood wave that exceeded the $HQ_{100}$ of 241 $m^3/s$ for the gauge Altenahr (Mohr et al., 2023) by far.

### 4.3 Downward counterfactual analysis for Germany

In this section we show the results of the downward counterfactual modelling for all subbasins in Germany. Because of the large amount of individual subbasins, spatial details cannot be shown. However, the results are also illustrated in a web application which allows zooming into regions of interest (Heistermann and Voit, 2023). Since larger subbasins can generate more runoff than smaller basins, we show the UPD (section 3.4) instead of the absolute peak discharge. On average, there are 41,873 counterfactuals for each subbasin. Figure 6a shows, for each subbasin, the highest UPD derived from original events (2001–2022) while Figures 6b and 6c show the maximum UPD and the 99th percentile of all counterfactual scenarios per subbasin.

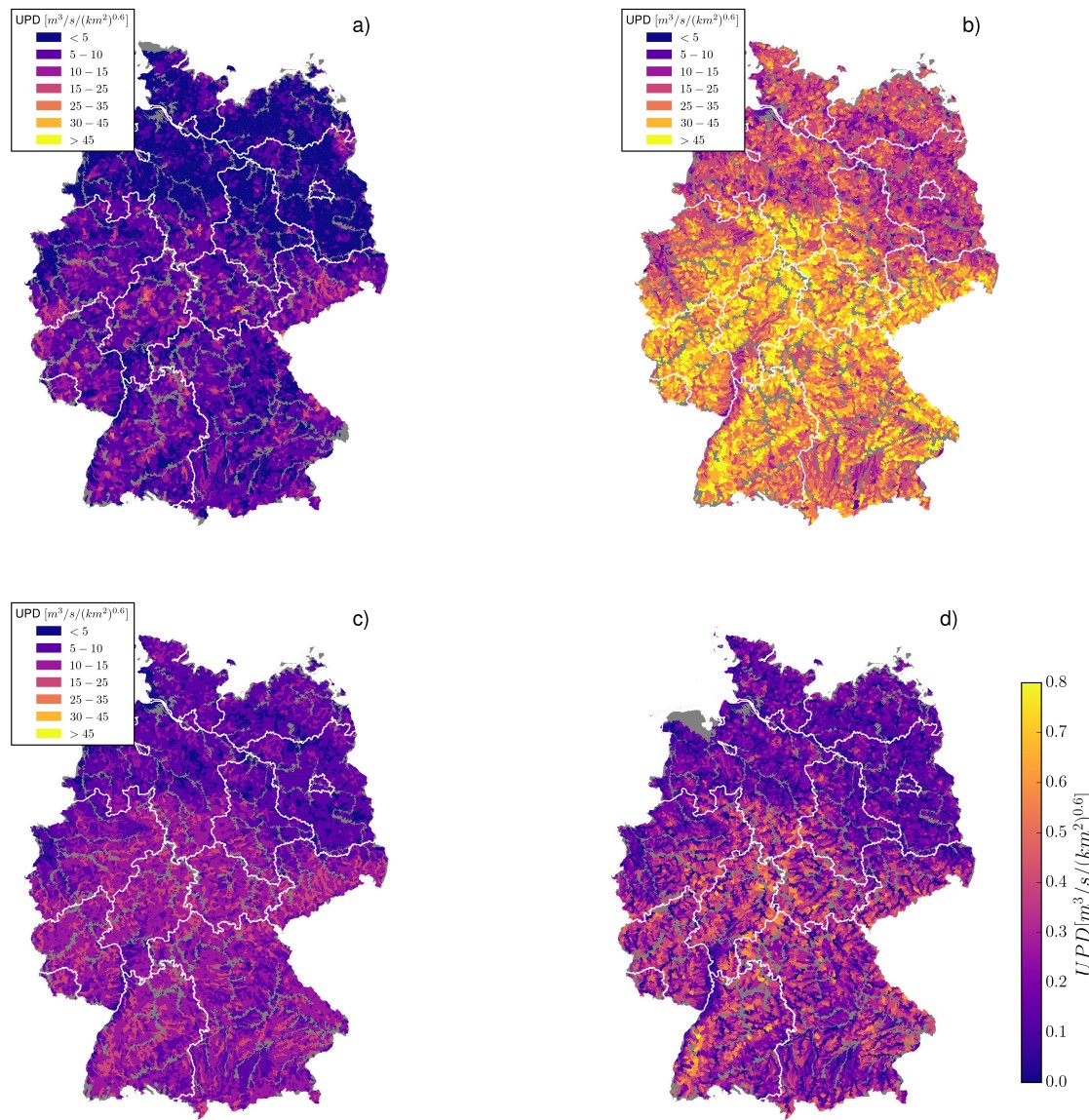

**Figure 6.** a) Maximum UPD from original events; b) Maximum counterfactual UPD; c) 99-percentile UPD derived from downward counterfactual simulations for Germany; d) shows the unit peak discharge derived only from the respective GIUHs. Grey: Basins with an area > 750 km$^2$ which were not considered in the analysis. White: federal state borders.

Looking at historical HPEs and consequent QR peaks that these events triggered, the downward counterfactual analysis is able to remove the random element of *where* an HPE occurred (Figures 6b, c). All but one basin showed much higher QR peaks

in response to downward counterfactual events than compared to QR peaks caused by original events (Tab. 3). Unsurprisingly,

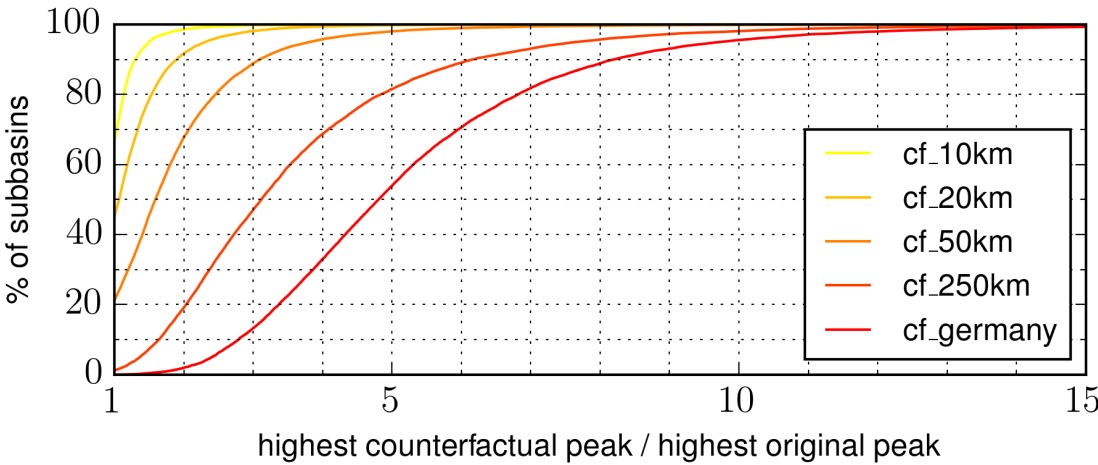

**Figure 7.** Cumulative distribution of the ratio between the highest counterfactual and the highest original peak for every subbasin is shown in red (cf_germany). From yellow to orange we show the same ratio but for counterfactuals with a limited shifting distance (10, 20, 50, 250 km, see section 3.3). QR peaks resulting from counterfactual simulations are much higher than the QR peaks caused by original events. As the shifting distance increases, more counterfactuals are considered for each subbasin. As a results, it becomes more likely that the counterfactual peaks are substantially higher than the highest original peak.

the distribution of the UPD in Germany is closely following the topography (Figures 6b, c and d). Mountain and low mountain ranges (compare to Fig. 1 display high QR peaks and therefore high UPD in the downward counterfactual analysis.

For headwater basins, where the QR peak does not depend on the inflow from any upstream basin, the GIUH can give a first idea of a basins's tendency for quick runoff concentration (Fig. 6 d). But contrary to the counterfactual simulations (Figures 6b and c), this does not give information about potential QR peak flow rates, yet. While GIUHs allow for a very efficient hydrological modelling and therefore make a downward counterfactual analysis possible, they use a uniform precipitation input. As shown in section 4.2 the spatial distribution of rainfall is highly important for the consequent QR peak. For this reason a detailed spatial resolution (a small subbasin size) is desirable to utilize radar rainfall data to its full extend. A small subbasin size consequently leads to a higher number of non-headwater basins whose QR peak characteristics can not be estimated with the GIUH.

Just for one single basin, the highest modelled peak was caused by an original event (which triggered a severe flash flood around Rudolstadt, Thuringia, on May 31st, 2008), in contrast to any counterfactual scenario. For 98 % of the basins, the downward counterfactual peak would be at least two times, for 47% at least five times higher than the highest observed peak in the last 22 years (Fig. 7). Figure 7 also shows the corresponding ratios for more "conservative" counterfactual scenarios for which the maximum shifting distance was limited to 10, 20, 50 or 250 km (see section 3.3). For the cf_50km scenario, for instance, 21 % of the discharge peaks from counterfactuals are not higher than the peaks caused by original events. This is due to the fact that a maximum shifting distance of 50 km will leave quite a number of subbasins essentially unaffected by

**Table 3.** Display of how often events (and their respective downward counterfactuals) caused the highest discharge in a subbasin (column "count"). "other" describes original events which do not belong to the top 10 events.

| Event | count |
|---|---|
| SN/Jun13 | 16248 |
| BB/Jun21 | 3033 |
| LS/Jul02 | 296 |
| SN/Aug02 | 227 |
| NW/Jul21 | 2 |
| BB/Jun20 | 2 |
| other | 1 |

the main footprint of the shifted HPE. Here we need to keep in mind that we only selected 10 out of 17,302 HPEs from the catalog for the counterfactual search. A better approach for designing such a conservative counterfactual search might be to select, for each subbasin, the most extreme HPE in a specific radius (say 50 km) and then shift this HPE over the corresponding subbasin. But even within the more conservative cf_50km dataset, 51 % of the basins exhibit a ratio of more than 1.5 between the counterfactual and the original peak; more than 30 % have a ratio of more than twice as high as the original peak. Especially in basins which have not yet been affected by severe flash floods in the recent past, the results from the counterfactual analysis could support the preparedness for flood events that might have been unexpected so far, based on observational records.

For the downward counterfactual study we shifted 10 extreme HPEs across Germany. Additionally, we modelled the runoff that was generated by all the HPEs in our catalog in their original spatial position. Table 3 shows which events caused the highest discharges for sub-basins all across Germany: the counterfactuals of the event SNJun/13 have caused the highest QR peaks in 82 % of the subbasins. Out of the ten HPEs, this is also the event with the highest hourly precipitation rates (see Tab. 1). Then again, the BB/Jun21 event also accounts for a substantial proportion of maximum counterfactual peaks while it only ranks sixth with regard to hourly precipitation levels. Only in two subbasins, the highest QR peaks were caused by NW/Jul21 counterfactuals. In only one case, the worst case scenario was caused by an original event. While we expect the maximum counterfactual peaks to be governed by the interaction of specific spatio-temporal HPE features and basin properties, the nature of this interaction remains yet to be explained. In other words, it should be subject to future research to better understand which features favour an exceptional runoff response at the flash-flood scale. Such research should not be limited to the top 10 events, but aim for a more comprehensive counterfactual search (see section 4.2).

The counterfactual analysis results in a large data set of potential QR peaks in Germany. Even though these QR peaks might not fully represent all processes involved in discharge generation they reflect the major runoff processes in small basins and show a range of plausible discharge cases which can be useful for further analysis. Specifically, the results could be used as a basis to further explore the geographic variation of the flash flood hazard in more detail, and to identify sub-basins that appear particularly prone to flash floods, mainly as a result of topographical controls.

## 5 Uncertainties and Limitations

In this section, we highlight the uncertainties and limitations that should be kept in mind when interpreting the above results.

### 5.1 Rainfall data and event catalog

Journée et al. (2023) showed that errors made by radar-based QPE are smaller than those obtained from rain gauge interpolation. Still, RADKLIM (Winterrath et al., 2018b) might considerably underestimate extreme precipitation. Such underestimation is typically caused by path-integrated attenuation effects (Jacobi and Heistermann, 2016), and it is not too uncommon that these effects are not sufficiently captured and corrected for by the applied rain-gauge adjustment methods (see e.g. Saadi et al. (2023)) for the NW/Jul21 event, or Bronstert et al. (2018) for the BW/May2016 event). Consequently, the resulting peak values of QR might be too low.

The same follows from the fact that the rainfall dataset, RADKLIM, is quite short from the perspective of extreme value statistics. While we argue that shifting HPEs across Germany might, to some extent, make up for this shortcoming, we have to prepare for that fact that other events are yet to be observed that might dwarf the top 10 events from our catalog.

And, finally, the top ten events from our catalog might not yet represent the worst case in terms of the QR response at the "flash flood scale". Particularly for very small headwater catchments, other events from the catalog could trigger higher runoff peaks even if their xWEI were smaller. For prospective research, other severity indices, ranking criteria or catalogs might still be considered or developed which could provide a more explicit focus on flash floods and might hence serve to an even more exhaustive counterfactual search.

Then again, the potential underestimation of rainfall also applies to the historical (original) events to which we compare the counterfactual events. Hence, the ratio between the historical and the maximum counterfactual peak flows might be more robust against any rainfall estimation bias – although we need to keep in mind the non-linear transformation of rainfall to runoff (see next section).

Some HPEs, e.g. the SN/Aug2002 or the NW/Jul21 event are not completely captured by the DWD's weather radar network, as they extended across the borders of Germany. For these events, the extremeness is necessarily underestimated. We still decided to use these HPEs in our counterfactual simulation experiment because they are, even while being incompletely captured, among the 10 most extreme HPEs observed in Germany within the last 22 years.

For DWD's operational radar-based precipitation product (RADOLAN), Saadi et al. (2023) reported an underestimation of 18% compared to rain gauges for the NW/Jul21 event; Bronstert et al. (2018) found an underestimation of about 30% for the BW/May2016 event. For the RADKLIM product, the uncertainty is expected to be lower than for the RADOLAN product, e.g. due to the usage of additional data for the rain-gauge adjustment. Yet, a systematic assessment of biases in RADKLIM is not yet available. In any case, the level of underestimation is expected to vary dramatically from event to event, as different sources of error govern the overall uncertainty in space and time (Heistermann et al., 2015).

## 5.2 Hydrological model

The applied hydrological model has, as any model, a number of limitations which we would like to discuss in more detail.

The unit hydrograph method assumes a linear and time-invariant response of a watershed to a spatially homogeneous pulse of effective rainfall (Yi et al., 2022). This assumption is a simplification.

The SCS-CN method implements antecedent soil moisture by considering the total rainfall amount within the last 5 days. Although we added a temporal buffer around our events, we always started the calculations assuming previously dry soils. While the modelled soil moisture class will change as the event unfolds, this assumption decreases runoff generation in the beginning of the event. The worst case scenario, in terms of QR peaks, would be saturated soils at the beginning of an event.

Since our model does not include base flow, there is certainly a small fraction of the total runoff missing in the QR peaks. Additionally, we know that clogging of bridges with uprooted trees and debris can play a major role in the formation of flood peaks (Borga et al., 2014). Neither does our model account for such effects, nor does it include a hydrodynamic channel model. Together with the expected underestimation of rainfall (see section 5.1) our results are likely to underestimate discharge peaks.

Utilizing a smaller subbasin size would be advantageous, particularly in the context of investigating flash floods. For example, within our chosen spatial discretization, we were unable to reproduce the extraordinary discharge peak during the Braunsbach flooding in May 2016 (Bronstert et al., 2018), which was generated in a subbasin of $6\,km^2$. However, computational cost increases exponentially with spatial resolution, so we did not implement smaller subbasins, yet.

Furthermore, our study relies on an uncalibrated model. The main reason for this is the lack of stream gauge records for small catchments. In addition, stream gauges are often unable to effectively observe extreme flash floods due to being damaged by the actual flood wave (Amponsah et al., 2018). Marchi et al. (2010) showed that only 20 % of flash flood events in small catchments were gauged by a stream gauge section. For these reasons, flash flood events are usually underrepresented in streamflow records (Borga et al., 2014). However, both model components, the SCS-CN model for QR formation and the GIUH for QR concentration are widely used and their applicability was validated in numerous contexts.

Taking all these aspects into account, we would like to emphasize once more that our model is not designed for precise discharge predictions. Instead, it serves as a tool to consistently represent the effects of rainfall, topography, soils and land use while enabling us to simulate a substantial number of counterfactual scenarios. This large number of simulations is a key feature of this study as it allows to comprehensively explore possible realisations of counterfactual rainfall events and their effect on peak discharge.

## 5.3 Spatial shifting of events

In our counterfactual analysis, we assumed that any of the analysed HPEs could have occurred anywhere in Germany. This is a very strong assumption, and it should be emphasized that the validity of this assumption remains an open question. Certainly, an HPE results from the interaction of large and regional scale circulation patterns with regional and local features of the earth's surface. E.g, orographic effects can augment precipitation and lead to anchoring convection (Marchi et al., 2010; Tarolli et al., 2013). Our study does not consider such effects, which could lead to unrealistic counterfactuals. For this reason we also carried

out a more conservative analysis in which we restricted the spatial shifting of HPEs to a radius of 10 km, 20 km, 50 km and 250 km around their original centroid. These results are displayed in Figure 7. It would also be very helpful for future research if the atmospheric modelling community further explored how exceptional HPEs could have unfolded under disturbed initial and boundary conditions, or under a warmer climate (see e.g. Ludwig et al., 2023a, for a pseudo global warming analysis of the July 2021 event), and thereby provide a better basis to evaluate the assumptions behind our counterfactual search.

## 6 Conclusions

In this study, we presented a downward counterfactual scenario analysis to assess the flash flood hazard in small to medium-sized basins in Germany. Instead of relying on local observational records of limited length, we identified the most severe heavy precipitation events from 2001 until 2022, and assumed that these events could have occurred anywhere in Germany. The quick runoff response to the resulting counterfactual rainfall scenarios was simulated by using a parsimonious and computationally

efficient rainfall-runoff model, and compared to the quick runoff response of historical events that actually took place in the corresponding subbasins.

By using a radar-based precipitation product, we were able to account in detail for the effects of different spatio-temporal event realisations on the quick runoff response. These effects can substantially moderate the role of the total accumulation of areal mean rainfall. This was first demonstrated in a case study of the July 2021 flood event (NW/Jul21) for the Ahr river

catchment, down to the runoff gauge Altenahr. Shifting the NW/Jul21 rainfall event in space resulted in a wide range of quick runoff peak values of which 6 percent exceeded the response to the original event. Furthermore, shifting another event (BB/Jun21), which had occurred one month earlier in eastern Germany, to the Ahr catchment effectuated a peak that exceeded the worst-case downward counterfactual peak of the NW/Jul21 by another 26 percent.

We then expanded the analysis to all of Germany and found that, on average, the worst case downward counterfactual

exceeded the maximum original quick runoff peak by a factor of 5.3. In general, the quick runoff response is dominated by topography. It turned out that the SN/Jun13 event (see Tab. 1) caused the maximum counterfactual peak in the majority of basins. Still, readers should be aware of various limitations of our approach which, some of which might lead to a considerable underestimation of counterfactual quick runoff peaks.

To make our results easily accessible, we created a web viewer where interested users can explore the results for each

subbasin in Germany (Heistermann and Voit, 2023). Still, our results leave various open questions: The most obvious, of course, is about the validity of shifting events all over Germany. Furthermore, focusing on the top 10 events as ranked by the xWEI might hide events that were more severe at the flash flood scale. So we should further explore the event catalog to understand which spatio-temporal structure makes an event particularly hazardous. Besides, it would be interesting to see how the counterfactual peaks compare to the values which are currently used for risk management. Furthermore, we just looked

at the worst case scenario for individual basins. However, large precipitation events can trigger flash floods in multiple basins simultaneously. The identification of regional flash flood clusters caused by one event are relevant in the context of disaster response. It should be clear that our design of counterfactual scenarios only addresses one single aspect: the spatial position of

the precipitation field and its effect on the hydrological hazard intensity. A more comprehensive counterfactual search would require accounting for impact-related aspects and processes. Such aspects could e.g. be the daytime or weekday at which an event occurs, the effectiveness of an early warning chain, or cascading effects of damages to infrastructure.

We would like to emphasize that the presented approach should be considered as a framework rather than a fixed method with fixed results: users could employ different catalogs, make different assumptions on spatial shifting of heavy precipitation events, use a different hydrological model, and define different metrics to assess the impact-relevance of the hydrological response. The key message here is that the presented framework for counterfactual scenario analysis provides a different view on flash flood hazards which should be helpful to reduce the element of surprise in disaster risk management.

*Code and data availability.* We published code and data to exemplify the computation of both *WEI* and *xWEI* in the following repository: https://doi.org/10.5281/zenodo.6556446. We publish notebooks and code which demonstrate our whole workflow for this study for a small, exemplary region (Altenahr basin, see section 4.2): the derivation of GIUHs from a digital elevation model, the extraction of rainfall data from and effective rainfall for the subbasins from RADKLIM data and the modelling of quick runoff. The code is published at: https://github.com/plvoit/counterfactual_flash_flood_analysis.

All data used in this study is accessible at the open data repository of the DWD: the RADKLIM_RW_2017.002 dataset is available at https://opendata.dwd.de/climate_environment/CDC/grids_germany/hourly/radolan/reproc/2017_002, (Winterrath et al., 2018b); the EU-DEM is available at https://ec.europa.eu/eurostat/web/gisco/geodata/reference-data/elevation/eu-dem/eu-dem-dd; the CLC5-2018 land cover data is available at https://gdz.bkg.bund.de/index.php/default/open-data/corine-land-cover-5-ha-stand-2018-clc5-2018.html. The soil data is available at https://www.bgr.bund.de/DE/Themen/Boden/Informationsgrundlagen/Bodenkundliche_Karten_Datenbanken/BUEK200/buek200_node.html, all last accessed 13 December 2023.

*Author contributions.* PV and MH conceptualized this study. PV developed the software and carried out the analysis; MH contributed to the analysis and developed the web viewer. PV prepared the manuscript with contributions of MH.

*Competing interests.* The contact author has declared that neither they nor their co-authors have any competing interests.

*Acknowledgements.* We would like to thank the open source community without its software and data this study would have not been possible. Some small parts of the text were improved in exchange with a language model (https://chat.openai.com/chat). We thank Boha Shehu for giving further insight into the reconstruction of discharge at the gauge Altenahr. Paul Voit was funded by the Deutsche Forschungsgemeinschaft (grant no. GRK 2043, project number 251036843).

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

## Appendix A:  Creation of HPE catalog

The catalog was created as follows (see also Fig. A1) for illustration). For simplification we just used only two durations in Figure A1 (1 and 72 h), while in our actual study we used eight durations (1, 2, 4, 6, 12, 24, 48, 72 h):

1. We applied a 3 km x 3 km x 72 km) moving window for each pixel in the RADKLIM dataset. In Figure A1 a) and b) the pixel is surrounded by a red box. In this moving window we aggregate the rainfall to the durations to respective durations (Figure A1 c) and d). For each duration we calculate the return periods for every pixel in the moving window (Figure A1 e and f). Now we can compute the xWEI. The return periods get sorted by decreasing order (Figure A1 g and h). We then compute the extremeness, $E_{tA}$ based on Müller and Kaspar (2014):

$$E_{tA} = \frac{\sum_{i=1}^{n} ln(p_{t,i})}{n} * \frac{\sqrt{A}}{\sqrt{\pi}} \qquad [ln(year)km] \tag{A1}$$

The process is explained in more detail in Voit and Heistermann (2022).

Following this procedure, we get an $E_{tA}$-curve for every duration (Fig. A1 i and j). The $E_{tA}$-curves are placed on a grid (Fig. A1 k). The $E_{tA}$-curves span a surface. The volume underneath that surface is the xWEI-value for the pixel (Fig. A1 l) which is high, if the rainfall in the 3 x 3 km neighborhood was extreme at multiple durations (between 1 h and 72 h).

2. This way the xWEI-moving window works as a filter for the rainfall data. The result is a dataset of xWEI values with the same dimensions (x, y, time) as the RADKLIM dataset. An xWEI value of ten is approximately equal to an event that had a return period of around 10 years on one duration and at a spatial scale of 9 km$^2$.

3. All cells with an xWEI < 10 were discarded (set to NaN) to ensure that there are just cells remaining which signify extreme rainfall. The remaining adjacent cells were clustered based on their neighborhood (pixels within 10 km). This way we obtained distinct clusters where the rainfall must have been exceptionally high.

4. Finally, we determined the bounding box and computed the xWEI value for the entire bounding box, for each identified cluster.

## Appendix B:  Distribution of subbasin sizes

Figure B1 shows the distribution of subbasin sizes for the study area Germany.

## Appendix C:  Description of Top 10 events

We supply further detail to the top ten events between 2001-2022 (section 4.1) which were identified using the procedure described in section 3.1 and A2.

- **LS/Jul02** hit the Harz mountains in the center of Germany with high rainfall sums and lead to flooding of some cities (e.g. Braunschweig). Apparently this HPE did not cause extensive damage as there is not much literature about this event, apart from local newspapers. Furthermore, this event was overshadowed by one of the largest flood catastrophes in Germany just one month later (SN/Aug2002). We can just hypothesize that the event would have caused more damages, had it not happened in the Harz area, which is a watershed. Additionally, there are large reservoirs in this area which regulate streamflow and might have prevented the formation of a larger flood wave.

- **BB/Jun17** caused massive urban flooding in Berlin. This HPE caused the largest insured losses in the period 2002 to 2017 (€60 million) in the greater Berlin area (Caldas-Alvarez et al., 2022).

- The **SN/Aug02** HPE caused extensive flooding in Central Europe (Germany, Austria, Czech Republic and Slovakia). The flooding occurred in the catchments of the Danube and the Elbe. In Germany alone the flood caused 21 casualties and a record breaking damage of €11.6 billion (Thieken et al., 2007; CRED/UCLouvain, 2023).

- Regarding damages, HPE **NW/Jul21** exceeded all previously recorded events even though the rainfall sums were not the most extreme, compared to other historic events (Ludwig et al., 2023b). The HPE affected mainly Belgium, the Netherlands and Western Germany. €40 billion damage and 191 casualties (CRED/UCLouvain, 2023) are the consequences of this HPEs.

- The flood following **LS/Jul17** caused damages in the districts surrounding the Harz mountains and the city of Hildesheim (Niedersächsischer Landesbetrieb für Wasserwirtschaft, Küsten- und Naturschutz (NLWKN), 2021). According to the DWD the meteorological extremeness of this HPE was similar to the infamous SN/Aug02 event, but due to the location the consequences were not as serious (Becker et al., 2017).

- **BW/May16** was a large HPE across Central Europe which affected Southern Germany. The event included episodes of intense small scale precipitation which caused e.g. the flash flood that partly destroyed the city of Braunsbach (Bronstert et al., 2018). This caused a damage of €2 billion. Euro and 7 deaths (CRED/UCLouvain, 2023).

- Even though **BB/Jun21** displayed the highest daily rainfall sum in Germany in 2021 (198.7 mm, Becker et al. (2017)) the event did not cause a lot of damage.

- The **BB/Jun20** HPE showed heavy rainfall, especially on shorter durations, in the Brandenburg area and caused smaller floods but did not cause extensive damages.

- Even though the precipitations sums during **HS/May19** exceeded a 100 year return period in many locations, this HPEs did not cause high damages.

- **SN/Jun13** hit central Europe and caused large-scale flooding of many rivers, mainly Danube and Elbe (Schröter et al., 2015). The event caused 4 casualties and an until then unseen damage of €12,9 billion (CRED/UCLouvain, 2023).

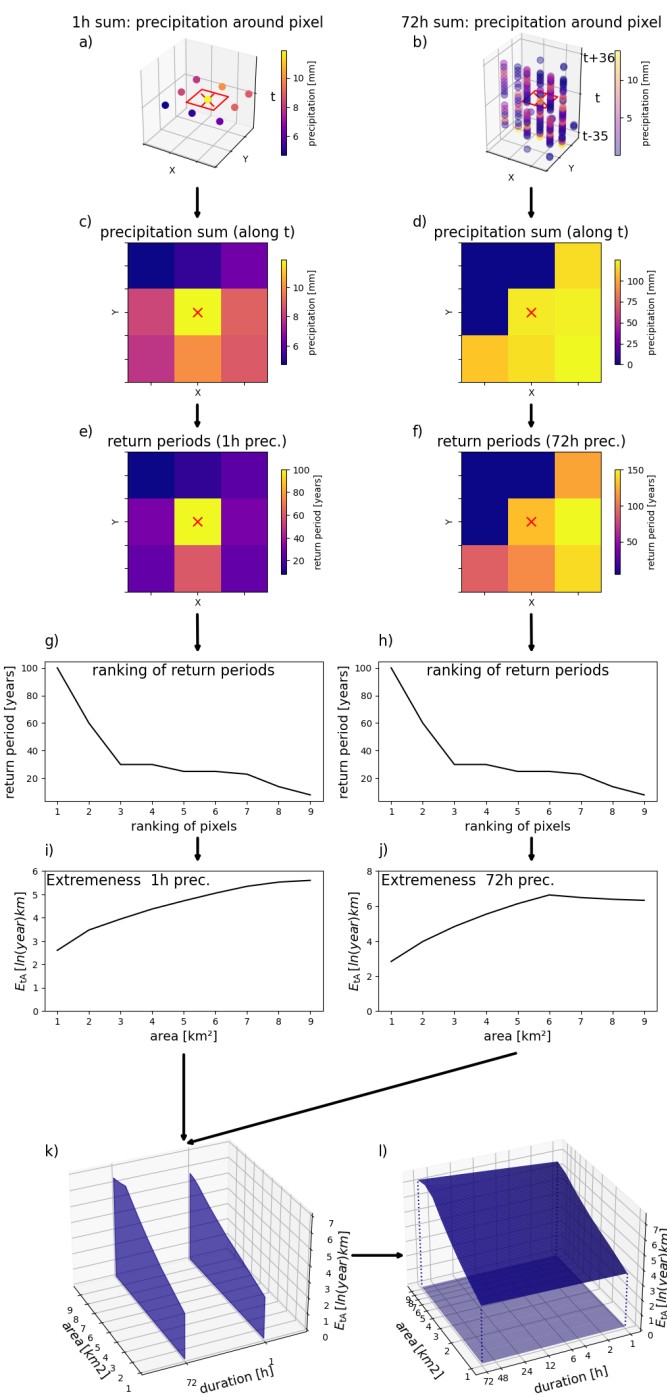

**Figure A1.** Pixel-wise computation of the xWEI: a) and b) the rainfall data in a 3x3 km neighborhood for the respective duration. a) 1h precipitation, b) 72h precipitation. c) and d) precipitation sums for the respective durations. e) and f): return periods of the precipitation sums. g) and h): ranked return periods. i) and j): $E_{tA}$-curves computed from the ranked return periods. k): The $E_{tA}$-curves are placed on a grid. l): a surface is spanned across the curves. The volume under this surface is the xWEI-value of the pixel.

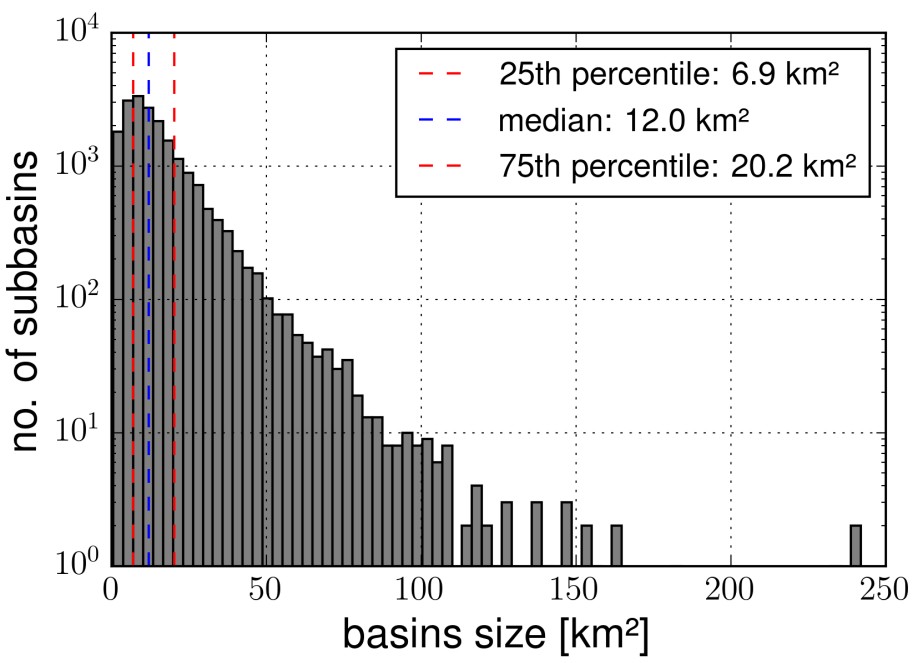

**Figure B1.** Distribution of subbasin sizes in the study area. The blue line indicates the median size, the red lines the 25- and 75-percentile.