# Peer review of "A downward counterfactual analysis of flash floods in Germany"

_Natural Hazards and Earth System Sciences, 2023_

## Referee Comment (RC1)

**Review nhess-2023-224**

Based on a radar-based precipitation dataset (2001 to 2022) of the German Weather Service, counterfactual studies on flash flood events in Germany are carried out. Counterfactual studies show what happens when an extreme event occurs under different conditions (e.g. at a different location) and can help disaster risk management to better prepare for such events. First, the ten most severe precipitation events in Germany are identified, which then serve as the basis for the counterfactual studies. In particular, the prominent case (floods in July 2021) is discussed in more detail. The TOP10 events are then shifted (depending on the distance from the point of origin) and the potential hydrological response to rare heavy precipitation events is analysed using a hydrological model. One aim is to determine how close the actual historical events are to the worst-case scenarios.

I find it a very nice paper with interesting results. However, these could be presented in a slightly more structured way, some detailed explanations for non-experts would be helpful and some of the highlights could be explained in more detail (especially part 2 of the results).

**Major:**

- Regarding our case study "Flood 2021/Ahr":
  Be careful... The flood event in July 2021 is a prominent example – but it is not a flash flood event (see title of your paper)? You need to discuss (or argue) this carefully.
- Section 4.1:
  Please discuss your TOP10 results in more detail. How severe were these events? Did these events have a relevant impact (damage, deaths)? Or were the events (or their effects) in the end not so bad? Please put your TOP10 in context with other work. Were these results also identified as "exceptional" by others? See also TOP-list by Ludwig et al. (2023): A multi-disciplinary analysis of the exceptional flood event of July 2021 in central Europe - Part 2: Historical context and relation to climate change, Nat. Hazards Earth Syst. Sci., 23, 1287-1311, https://doi.org/10.5194/nhess-23-1287-2023, 2023.
- In some passages, I would wish more context for readers who are not familiar with the topic (or from other disciplines). (More complex) figures could be better introduced: what is the motivation of the figure or what exactly does the figure show before starting the interpretation (e.g. Fig 3, 4, 6).
  In addition, the authors do not take international readers into account – for them a figure with relevant names (cities, mountains) would be very helpful; this figure should also contain the orography that is relevant for the interpretation of Figure 5.
- Please reflect your structure of the section 4. I had to read it twice to understand it. Please try to introduce everything before you start the interpretation (e.g. introduce complex figures before you discuss them). A clear sentence at the beginning (motivation for the following section, aim or scientific question to be investigated) could help the reader to understand more quickly what you are trying to achieve. You have nice results, but could discuss them in a more structured way.
- L321f: "The dataset can also be an helpful asset to identify flash flood hot spots in Germany."
  And what exactly would the hot spots be? Somehow the result is missing? Please work out better which regions would be particularly affected. At the moment you remain rather abstract. Maybe you should think about reducing the high resolution of Figure 5? Would a figure based on the "factors" be an idea (to show which postcode areas have been "too little" affected so far or where the danger could currently be underestimated too much due to the "missing" history)?

**Minor comments:**

L68f: Suggestion: Here you can again clearly define or clarify the study area for the reader.

Sect. 2.1. Prec data
Which season? The whole year or just the summer half-year?

L105ff: What are the advantages/disadvantages of xWEI compared to CatRaRE; again emphasise in writing exactly what is considered in xWEI that is not considered in CatRaRE (soil moisture is indirectly implied but clearly communicated).

L117: "...high, if the rainfall in the 3 km x 3 km neighborhood was extreme at multiple durations (between 1 h and 72 h)."
Perhaps add an example to make it easier for the reader to understand?

L128: "The study area is Germany."
Suggestion: Define this already at the beginning of section 2.

L1334: "The median basin size is 11.55 km2."
Idea/Suggestion: Add a histogram of all catchments; including quartile value (25%/75% percentile).

L135: "case study 1"
Do you have one case study or more? (because you write "1")

L167 "SCS-CN" method
Reference? References for "established"

L206: Suggestion: Include the table in results section 4 (or 4.1) (together with the spatial results) and only refer to this section here.

L209: "within the HPE's bounding box (not just for the aligned subbasin)".
Can this  be better explained what is meant by this?

L212: "By following this procedure, we generated approximately 230,000 QR data sets across Germany of counterfactual run off data sets."
Can the calculation of the value 230,000 be better explained to the reader?

Table 1: "The ID was constructed from an acronym that specifies the federal state"
A non-German wouldn't understand this, let alone that the names of the cities mean anything to them; could the spatial allocation be better explained to the international reader?

Table1: Ranking (1, 2, ....) is not given; not all dates have been transformed into English.

L216: "unit peak discharge (UPD)"
Reference?

L228f: "events with a large bounding box"
Please specify what you mean by this.

L229: "a large xWEI are likely to include smaller event clusters"
Do you mean: "a large xWEI **value** are likely to include **some** smaller event clusters"?

L241/Figure 3: Please introduce the figure better in the main text (more explanation) before going into the result (as the figure is quite complex).

L247: "the QR peak can vary by a factor of 2."
Can you give a reason (or hypothesis) why this is the case?

Figure 4: Can you explain one or two sentences about "stream order" for the reader who is not so familiar with hydrology?

L254: Location of NWJul21_b events? How far away?

258f: "In our model, the catchment consist of 37 subbasins (Fig. 2a)."
This information would be helpful to the reader before.

L259f: "Using the same approach, we are also able to take the other nine events from Table 1 and compare the resulting QR flood peaks"
Please be more specific; you mean that you apply the other observed cases to the Ahr area to see that...

L263f: "have apparently caused even higher QR peaks in Altenahr"
Can you hypothesize why this is the case?

L266f: "Among these, there are no counterfactuals of the events BW/May16, BB/Jun17, LS/Jul17, HS/May19 and BB/Jun20"
Why?

L288f: "Mountain and low mountain ranges such as the Harz mountains, Ore mountains, the Black Forest"
International readers need a figure that contains relevant location information; this could also include the orography so that the relationship is clear to them.

Figure 6: I don't understand the figure; please provide a more precise caption. A caption should first explain the figure (in a scientific paper). You start with results.

Table 3 & LL314ff: "Table 3 shows which events caused the highest discharges for sub-basins all across Germany: the counterfactuals of the event SNJun/13 have caused the highest QR peaks in 82 % of the subbasins. Out of the ten HPEs, this is also the event with the highest hourly precipitation rates (see Tab. 1). Only in two subbasins, the highest QR peaks were caused by NW/Jul21 counterfactuals. In only one case, the worst case scenario was caused by an original event."
And what does that mean; can you elaborate on the highlight of this statement? Can you be more specific? What do we learn from this finding?

Section 5:
A brief  introduction and motivation for the section would be helpful to the reader.
Perhaps rename section 5 to "Discussion" or "Discussion and Limitations"?

L342ff: This section seems misplaced here; suggest moving it to 4.1.

L356: "Together with the expected underestimation of rainfall (see section 5.1)"
What order of magnitude are we talking about here?

---

## Author Comment (AC2)

**Interactive Discussion: Author Response to Referee #2**

**A downward counterfactual analysis of flash floods in Germany**

Paul Voit and Maik Heistermann

*NHESS Discussions,* `doi:10.5194/nhess-2023-224`
* * *
**RC:** *Reviewer Comment*,     AR: *Author Response*,     ☐ Manuscript text

Dear Referee,

thank you for accepting to review this paper. We are very glad about the positive feedback and about your constructive suggestions.

Please find our responses to your comments below. These should be considered as preliminary (part of the interactive discussion). The final implementation of changes also depends on another referee report.

Thanks again for your efforts!

Kind regards,
Paul Voit and Maik Heistermann

**RC:** *Abstract: You should define the concept of counterfactual thinking in the abstract, for non-experts.*

AR: We inserted a brief explanation after the first sentence of the abstract:

> Counterfactuals are scenarios that describe alternative ways of how an event, in this case an extreme rainfall event, could have unfolded.

**RC:** *Line 72 : It would be good to clearly define the time window analysed (e.g. by just adding "between 2001 and 2022" at the end of line 72).*

AR: Thank you, indeed, this information is missing there. We added it, as suggested at the end of line 72 of the preprint.

> To allow for a detailed representation of the spatio-temporal variability of rainfall, we used the radar climatology product (RADKLIM v2017.002) provided by Germany's national meteorological service (Deutscher Wetterdienst; DWD hereafter) between 2001 and 2022.

**RC:** *Line 113: It might be difficult to do, but I think that a figure illustrating the method and its different stages would make it much easier to understand.*

AR: This point was also mentioned by Referee #1. We will add a plot which hopefully explains the process a bit better to the reader. As this Figure, however, is quite bulky, and as not all readers might be interested in this level of technical detail, we suggest to shorten, in the main article, the explanation of how the catalog was generated. Instead, we provide a more detailed explanation, together with the new figure, in the supplementary material and refer to this in the main text.

Hence, we replaced ll. 113-126 of the preprint by the following shorter text:

> The catalog was created by applying multi-step procedure. If we consider the RADKLIM dataset as a 3-D array (one temporal dimension, two spatial dimensions), we first apply a moving 3-D window (72 hours x 3 km x 3 km) to the entire dataset. Within this moving window, the rainfall extremeness is computed for each voxel and for various durations. Afterwards, a clustering algorithm is applied to identify spatio-temporal clusters of extreme rainfall. The details of this approach together with an illustration are provided in the supplementary material.

In the supplementary, we suggest to use the following Figure 1 for illustrating the process (you can find the figure at the end this PDF because it is quite large):

Referring to this figure, we will explain the process in the supplementary as follows:

The catalog was created as follows (see also Fig. S1) for illustration). For simplification we just used only two durations in Figure S1 (1 and 72 h), while in our actual study we used eight durations (1, 2, 4, 6, 12, 24, 48, 72 h):

1. We applied a 3 km x 3 km x 72 km) moving window for each pixel in the RADKLIM dataset. In Figure S1 a) and b) the pixel is surrounded by a red box. In this moving window we aggregate the rainfall to the durations to respective durations (Figure S1 c) and d). For each duration we calculate the return periods for every pixel in the moving window (Figure S1 e) and f)). Now we can compute the xWEI. The return periods get sorted by decreasing order (Figure S1 g) and h). We then compute the extremeness, $E_{tA}$ based on Müller et al., 2014:

$$E_{tA} = \frac{\sum_{i=1}^{n} ln(p_{t,i})}{n} * \frac{\sqrt{A}}{\sqrt{\pi}} \qquad [ln(year)km] \qquad (1)$$

The process is explained in more detail in Voit and Heistermann (2022).

Following this procedure, we get an $E_{tA}$-curve for every duration (Fig. S1 i) and j)). The $E_{tA}$-curves are placed on a grid (Fig. S1 k)). The $E_{tA}$-curves span a surface. The volume underneath that surface is the xWEI-value for the pixel (Fig. S1 l)) which is high, if the rainfall in the 3 x 3 km neighborhood was extreme at multiple durations (between 1 h and 72 h).

2. This way the xWEI-moving window works as a filter for the rainfall data. The result is a dataset of xWEI values with the same dimensions (x, y, time) as the RADKLIM dataset. An xWEI value of ten is approximately equal to an event that had a return period of around 10 years on one duration and at a spatial scale of 9 km².

3. All cells with an xWEI < 10 were discarded (set to NaN) to ensure that there are just cells remaining which signify extreme rainfall. The remaining adjacent cells were clustered based on their neighborhood (pixels within 10 km). This way we obtained distinct clusters where the rainfall must have been exceptionally high.

4. Finally, we determined the bounding box and computed the xWEI value for the entire bounding box, for each identified cluster.

**RC:** *Line 116 : Maybe you could briefly explain how is calculated the xWEI, and give some orders of magnitude.*

AR: We refer to the previous answer which now contains more details about the computation of the xWEI.

**RC:** *Line 123: Can you detail "Clustered based on their neighborhood" ? Or show it on an example if you decide to include an illustration.*

AR: We cluster pixels with high xWEI values in a neighborhood of 10 km (voxel-based clustering). Also this is now included in the previous explanation.

**RC:** *Line 133: Can you explain why the upper limit is precisely 750 km² for the catchment size? Could you give the minimum catchment size?*

AR: The basin size for typical flash floods varies among different authors:

- Marchi et al. (2010) and Charpentier-Noyer et al. (2023) define the typical spatial scale "less than

1000 km²".

- Gaume et al. (2008) refers to a value of 500 km² which has also been used by Matthai (1969) and Stǎnescu et al. (2004).

- Amponsah et al. (2018) state "catchment scales impacted by flash floods are generally less than 2000–3000 km2 in size".

Based on these different values we picked a rather small upper limit of 750 km² due to our simplistic model.

Regarding the distribution of basin sizes, we will add Figure 2 in the supplementary material.

**RC:** *Table 1: Adding information about impacts (e.g. damage) would help understanding the gravity of these events, which are not well-known by international readers.*

AR: Also referee #1 was requesting more information about the events. We included detailed explanation in the supplementary material:

- **LS/Jul02** hit the Harz mountains in the center of Germany with high rainfall sums and lead to flooding of some cities (e.g. Braunschweig). Apparently this HPE did not cause extensive damage as there is not much literature about this event, apart from local newspapers. Furthermore, this event was overshadowed by one of the largest flood catastrophes in Germany just one month later (SN/Aug2002). We can just hypothesize that the event would have caused more damages, had it not happened in the Harz area, which is a watershed. Additionally, there a large reservoirs in this area which regulate streamflow and might have prevented the formation of a larger flood wave.

- **BB/Jun17** caused massive urban flooding in Berlin. This HPE caused the largest insured losses in the period 2002 to 2017 (€60 million) in the greater Berlin area (Caldas-Alvarez et al., 2022).

- The **SN/Aug02** HPE caused extensive flooding in Central Europe (Germany, Austria, Czech Republic and Slovakia). The flooding occurred in the catchments of the Danube and the Elbe. In Germany alone the flood caused 21 casualties and a record breaking damage of €11.6 billion (Thieken et al., 2007; CRED/UCLouvain, 2023).

- Regarding damages, HPE **NW/Jul21** exceeded all previously recorded events even though the rainfall sums were not the most extreme, compared to other historic events (Ludwig et al., 2023). The HPE affected mainly Belgium, the Netherlands and Western Germany. €40 billion damage and 191 casualities (CRED/UCLouvain, 2023) are the consequences of this HPEs.

- The flood following **LS/Jul17** caused damages in the districts surrounding the Harz mountains and the city of Hildesheim (Niedersächsischer Landesbetrieb für Wasserwirtschaft, Küsten- und Naturschutz (NLWKN), 2021). According to the DWD the meteorological extremeness of this HPE was similar to the infamous SN/Aug02 event, but due to the location the consequences were not as serious (Becker et al., 2017).

- **BW/May16** was a large HPE across Central Europe which affected Southern Germany. The event included episodes of intense small scale precipitation which caused e.g. the flash flood that partly destroyed the city of Braunsbach (Bronstert et al., 2018). The caused a damage of €2 billion. Euro and 7 deaths (CRED/UCLouvain, 2023).

- Even though **BB/Jun21** displayed the highest daily rainfall sum in Germany in 2021 (198.7 mm, (Becker et al., 2017)) the event did not cause a lot of damage.

- The **BB/Jun20** HPE showed heavy rainfall, especially on shorter durations, in the Brandenburg area and caused smaller floods but did not cause extensive damages.

- Even though the precipitations sums during **HS/May19** exceeded a 100 year return period in many locations, this HPEs did not cause high damages.

- **SN/Jun13** hit central Europe and caused large-scale flooding of many rivers, mainly Danube and Elbe (Schröter et al., 2015). The event caused 4 casualties and an until then unseen damage of €12,9 billion (CRED/UCLouvain, 2023).

And this is the paragraph we will add to the main text after line 231 of the preprint:

> Very different levels of impacts were reported for these events. In section S2 of the supplementary material, we put each event in context to other available references (scientific or media), and also attempt to compile estimates of reported damages and loss of lives, if available. While all ten events featured exceptional amounts of rainfall and a corresponding runoff response, only five of them caused massive impacts (SN/Aug02, SN/Jun13, BW/May16, BB/Jun17, and, with by far the highest impact, NW/Jul21) while for the remaining events (LS/Jul02, LS/Jul17, LS/Jul17, HS/May19, and BB/Jun21), the impact was apparently not high enough to attract attention beyond the affected regions. The results of the counterfactual scenario analysis, as presented in the following, should help to understand whether the different levels of impacts for these events were mainly caused by their specific geographic position.

**RC:** *Figure 5: Could the legend be standardised?*

AR: Thank you for this good suggestion. A standardised legend would indeed make subplots a), b) and c) more informative and comparable. We therefore followed your suggestion (Figure 3 in this response letter).

**RC:** *Section 5/ Section 6: In my opinion it would be interesting to discuss the fact that this counterfactual approach is "only" performed from a hydrological point of view (hazard-based). However if you want to go to the bottom of the question "how close actual historical events have already touched upon the worst case scenario", you would need to shift to an impact-based approach.*

AR: We agree that such an impact-based perspective is important. We therefore added the following sentence in line 410

> It should be clear that our design of counterfactual scenarios only addresses one single aspect: the spatial position of the precipitation field and its effect on the hydrological hazard intensity. A more comprehensive counterfactual search would require accounting for impact-related aspects and processes. Such aspects could e.g. be the daytime or weekday at which an event occurs, the effectiveness of an early warning chain, or cascading effects of damages to infrastructure.

**References**

Amponsah, W., Ayral, P.-A., Boudevillain, B., Bouvier, C., Braud, I., Brunet, P., Delrieu, G., Didon-Lescot, J.-F., Gaume, E., Lebouc, L., et al.: Integrated high-resolution dataset of high-intensity European and Mediterranean flash floods, Earth System Science Data, 10, 1783–1794, 10.5194/nhess-2023-22410.5194/essd-10-1783-2018, 2018.

Becker, A., Junghänel, T., Hafer, M., Köcher, A., Rustemeier, E., Weigl, E., and Wittich, K.-P.: 2017, Juli: Einordnung der Stark- und Dauerregen in Deutschland zum Ende eines sehr nassen Juli 2017, URL https://www.dwd.de/DE/klimaumwelt/klimawandel/_functions/aktuellemeldungen/170731_starkniederschlaege_einordnung.html, 2017.

Bronstert, A., Agarwal, A., Boessenkool, B., Crisologo, I., Fischer, M., Heistermann, M., Köhn-Reich, L., López-Tarazón, J. A., Moran, T., Ozturk, U., et al.: Forensic hydro-meteorological analysis of an extreme flash flood: The 2016-05-29 event in Braunsbach, SW Germany, Science of the total environment, 630, 977–991, 10.5194/nhess-2023-22410.1016/j.scitotenv.2018.02.241, 2018.

Caldas-Alvarez, A., Augenstein, M., Ayzel, G., Barfus, K., Cherian, R., Dillenardt, L., Fauer, F., Feldmann, H., Heistermann, M., Karwat, A., Kaspar, F., Kreibich, H., Lucio-Eceiza, E. E., Meredith, E. P., Mohr, S., Niermann, D., Pfahl, S., Ruff, F., Rust, H. W., Schoppa, L., Schwitalla, T., Steidl, S., Thieken, A. H., Tradowsky, J. S., Wulfmeyer, V., and Quaas, J.: Meteorological, impact and climate perspectives of the 29 June 2017 heavy precipitation event in the Berlin metropolitan area, Natural Hazards and Earth System Sciences, 22, 3701–3724, 10.5194/nhess-2023-22410.5194/nhess-22-3701-2022, 2022.

Charpentier-Noyer, M., Peredo, D., Fleury, A., Marchal, H., Bouttier, F., Gaume, E., Nicolle, P., Payrastre, O., and Ramos, M.-H.: A methodological framework for the evaluation of short-range flash-flood hydrometeorological forecasts at the event scale, Natural Hazards and Earth System Sciences, 23, 2001–2029, 10.5194/nhess-2023-22410.5194/nhess-23-2001-2023, 2023.

CRED/UCLouvain: EM-DAT International Disaster Databse, URL www.emdat.be, 2023.

Gaume, E., Bain, V., Bernardara, P., Newinger, O., Barbuc, M., Bateman, A., Blašković ová, L., Blöschl, G., Borga, M., Dumitrescu, A., et al.: A compilation of data on European flash floods, Journal of Hydrology, 367, 70–78, 10.5194/nhess-2023-22410.1016/j.jhydrol.2008.12.028, 2008.

Ludwig, P., Ehmele, F., Franca, M. J., Mohr, S., Caldas-Alvarez, A., Daniell, J. E., Ehret, U., Feldmann, H., Hundhausen, M., Knippertz, P., Küpfer, K., Kunz, M., Mühr, B., Pinto, J. G., Quinting, J., Schäfer, A. M., Seidel, F., and Wisotzky, C.: A multi-disciplinary analysis of the exceptional flood event of July 2021 in central Europe – Part 2: Historical context and relation to climate change, Natural Hazards and Earth System Sciences, 23, 1287–1311, 10.5194/nhess-2023-22410.5194/nhess-23-1287-2023, 2023.

Marchi, L., Borga, M., Preciso, E., and Gaume, E.: Characterisation of selected extreme flash floods in Europe and implications for flood risk management, Journal of Hydrology, 394, 118–133, 10.5194/nhess-2023-22410.1016/j.jhydrol.2010.07.017, 2010.

Matthai, H. F.: Floods of June 1965 in South Platte River basin, Colorado, US Government Printing Office, 1969.

Niedersächsischer Landesbetrieb für Wasserwirtschaft, Küsten- und Naturschutz (NLWKN): Das Juli-Hochwasser 2017 im südlichen Niedersachsen, URL https://www.nlwkn.niedersachsen.de/download/124949, 2021.

Schröter, K., Kunz, M., Elmer, F., Mühr, B., and Merz, B.: What made the June 2013 flood in Germany an exceptional event? A hydro-meteorological evaluation, Hydrology and Earth System Sciences, 19, 309–327, 10.5194/nhess-2023-224/10.5194/hess-19-309-2015, publisher: Copernicus GmbH, 2015.

Stănescu, V., Ungureanu, V., and Mătreaţă, M.: Regional analysis of annual peak discharges in the Danube catchment (follow-up volume no. VII to the Danube Monograph), INMH, Bucharest, 2004.

Thieken, A. H., Kreibich, H., Müller, M., and Merz, B.: Coping with floods: preparedness, response and recovery of flood-affected residents in Germany in 2002, Hydrological Sciences Journal, 52, 1016–1037, 2007.

Voit, P. and Heistermann, M.: A new index to quantify the extremeness of precipitation across scales, Natural Hazards and Earth System Sciences, 22, 2791–2805, 10.5194/nhess-2023-22410.5194/nhess-22-2791-2022, 2022.

[Figure]

Figure 1: Pixel-wise computation of the xWEI: a) and b) the rainfall data in 3x3 km neighborhood of the for the respective duration. a) 1h precipitation, b) 72h precipitation. c) and d) precipitation sums for the respective durations. e) and f): return periods of the precipitation sums. g) and h): ranked return periods. i) and j): $E_{tA}$-curves computed from the ranked return periods. k): The $E_{tA}$-curves are placed on a grid. l): a surface is spanned across the curves. The volume under this surface is the xWEI-value of the pixel.

[Figure]

Figure 2: Distribution of subbasin sizes in the study area. The blue line indicates the median size, the red lines the 25- and 75-percentile.

[Figure]

Figure 3: a) Maximum UPD from original events; b) Maximum counterfactual UPD derived from downward counterfactual simulations for Germany; c) 75-percentile UPD derived from downward counterfactual simulations for Germany; d) shows the unit peak discharge derived only from the respective GIUHs. Grey: Basins with an area > 750 km² which were not considered in the analysis. White: federal state borders.

---

## Author Comment (AC3)

**Interactive Discussion: Author Response to Referee #1**

**A downward counterfactual analysis of flash floods in Germany**

Paul Voit and Maik Heistermann

*NHESS Discussions,* `doi:10.5194/nhess-2023-224`
* * *
**RC:** *Reviewer Comment*,     AR: *Author Response*,     ☐ Manuscript text

Dear Referee,

thank you for your thorough and constructive feedback and the effort to review this manuscript. Surely, incorporating your suggestions will improve the readability and overall quality of the manuscript.

Please find our responses to your comments below. These should be considered as preliminary (part of the interactive discussion). The final implementation of changes also depends on another referee report.

Thanks again for your efforts!

Kind regards,
Paul Voit and Maik Heistermann

**RC:** *I find it a very nice paper with interesting results. However, these could be presented in a slightly more structured way, some detailed explanations for non-experts would be helpful and some of the highlights could be explained in more detail (especially part 2 of the results).*

AR: We agree that in some parts of the manuscript, a more detailed explanation will increase the comprehensibility of the paper for various audiences. At the same time, we need to keep the focus of the manuscript to the main line of thought. So while we added more explanation, more "guidance" through the manuscript, we decided, for some of the referees suggestions, to add the additional information to a supplementary material. This way, we attempt to provide a balance between detail and reference. In the following part of this response letter, we will always document which new information will be part of the actual manuscript and which will be included in the supplementary.

**RC:** *Regarding our case study "Flood 2021/Ahr": Be careful... The flood event in July 2021 is a prominent example – but it is not a flash flood event (see title of your paper)? You need to discuss (or argue) this carefully.*

AR: Thank you for pointing out this issue. In fact, the flood peak arrived at Altenahr about eight hours after the centroid of the effective rainfall (according to our model, see Fig. 1 in this response letter). According to the most common definitions, a flash flood has a lag time of up to six hours between the flood peak and the centroid of the effective rainfall (Marchi et al., 2010, Morin et al., 2002). So the flood event down to Altenahr misses this criterion by two hours.

[Figure]

Figure 1: Quick runoff (blue line) and areal average effective rainfall (blue bars) for the Altenahr basin. Red dotted lines: centroid of effective rainfall and QR peak

Still, several sub-basins within the Altenahr catchment show a very swift runoff peak within less than 6 hours. 22 out 23 headwater catchments even have a lag time of less than three hours.

We picked this event as a case study for various reasons: of course due to its significance in the discussion of flood risk management, but also because it shows key features of a flash flood, even if it has a lag time slightly longer than six hours. Furthermore, it is a very illustrative example of an event that is extreme across spatio-temporal scales, so that a superposition of flash floods at the headwater scale led to a massive runoff response at the meso-scale.

Given all these reasons, we fully agree with the referee that the choice of this case study warrants a careful justification to the audience. We therefore suggest to add an explanation at the beginning of section 4.2, after line 235 of the preprint:

> Typically, a flash flood is characterized by a lag time of up to six hours between the centroid of the effective rainfall and the hydrograph peak (Borga et. al., 2008; Marchi et al. 2010, Morin et al., 2002). So, strictly speaking, the flood event at Altenahr does not qualify as a flash flood: according to our model, the lag time at Altenahr amounted to approximately eight hours. Still, the event at Altenahr is a highly illustrative example for a swift and massive runoff response at the meso-scale which is the result of the temporal superposition of various upstream flash floods. In fact, all 23 sub-basins upstream of Altenahr show a lag time of less than 6 hours, 22 of them even less than 3 hours.

**RC:** *Section 4.1: Please discuss your TOP10 results in more detail. How severe were these events? Did these events have a relevant impact (damage, deaths)? Or were the events (or their effects) in the end not so bad?*

*Please put your TOP10 in context with other work. Were these results also identified as "exceptional" by others? See also TOP-list by Ludwig et al. (2023): A multi-disciplinary analysis of the exceptional flood event of July 2021 in central Europe - Part 2: Historical context and relation to climate change, Nat. Hazards Earth Syst. Sci., 23, 1287-1311, https://doi.org/10.5194/nhess-23-1287-2023, 2023.*

AR: We agree that more context for the selected events will be helpful for the readers. Thank you for providing the reference of Ludwig et al. (2023). It is in fact interesting and relevant to see that some of these events brought substantial impacts (of course the NW/Jul21 event, but also SN/Jun13, SN/Aug02, BW/May16, and BB/Jun17) while the impacts of others were measurable, but apparently not high enough to generate nation-wide media coverage and documentation in impact databases. This is an important feature of our approach: to find events that might have caused massive impacts if they had occurred somewhere else. The BB/Jun21 event is maybe the most illustrative example for an event that had negligible impacts, but would have caused a disaster elsewhere.

But while we agree that this kind of information is relevant and worthwhile, we think that a comprehensive documentation of context and impacts for each of the ten events would take away the focus too much and unnecessarily inflate the paper. We hence suggest the following solution: We will add a sub-section to a supplementary in which we will put each event in context to other references, and provide information on impacts, if available. We will refer to this supplementary in the main text. Furthermore, we will also add a short paragraph to the main text in which we will summarize this information (as discussed above). We will add this additional paragraph after line 231.

This is an outline of the list that we will add to the supplementary:

- **LS/Jul02** hit the Harz mountains in the center of Germany with high rainfall sums and lead to flooding of some cities (e.g. Braunschweig). Apparently this HPE did not cause extensive damage as there is not much literature about this event, apart from local newspapers. Furthermore, this event was overshadowed by one of the largest flood catastrophes in Germany just one month later (SN/Aug2002). We can just hypothesize that the event would have caused more damages, had it not happened in the Harz area, which is a watershed. Additionally, there a large reservoirs in this area which regulate streamflow and might have prevented the formation of a larger flood wave.

- **BB/Jun17** caused massive urban flooding in Berlin. This HPE caused the largest insured losses in the period 2002 to 2017 (€60 million) in the greater Berlin area (Caldas-Alvarez et al., 2022).

- The **SN/Aug02** HPE caused extensive flooding in Central Europe (Germany, Austria, Czech Republic and Slovakia). The flooding occurred in the catchments of the Danube and the Elbe. In Germany alone the flood caused 21 casualties and a record breaking damage of €11.6 billion (Thieken et al., 2007; CRED/UCLouvain, 2023).

- Regarding damages, HPE **NW/Jul21** exceeded all previously recorded events even though the rainfall sums were not the most extreme, compared to other historic events (Ludwig et al., 2023). The HPE affected mainly Belgium, the Netherlands and Western Germany. €40 billion damage and 191 casualities (CRED/UCLouvain, 2023) are the consequences of this HPEs.

- The flood following **LS/Jul17** caused damages in the districts surrounding the Harz mountains and the city of Hildesheim (Niedersächsischer Landesbetrieb für Wasserwirtschaft, Küsten- und Naturschutz (NLWKN), 2021). According to the DWD the meteorological extremeness of this HPE was similar to the infamous SN/Aug02 event, but due to the location the consequences were not as serious (Becker et al., 2017).

- **BW/May16** was a large HPE across Central Europe which affected Southern Germany. The event included episodes of intense small scale precipitation which caused e.g. the flash flood that partly destroyed the city of Braunsbach (Bronstert et al., 2018). The caused a damage of €2 billion. Euro and 7 deaths (CRED/UCLouvain, 2023).

- Even though **BB/Jun21** displayed the highest daily rainfall sum in Germany in 2021 (198.7 mm, (Becker et al., 2017)) the event did not cause a lot of damage.

- The **BB/Jun20** HPE showed heavy rainfall, especially on shorter durations, in the Brandenburg area and caused smaller floods but did not cause extensive damages.

- Even though the precipitations sums during **HS/May19** exceeded a 100 year return period in many locations, this HPEs did not cause high damages.

- **SN/Jun13** hit central Europe and caused large-scale flooding of many rivers, mainly Danube and Elbe (Schröter et al., 2015). The event caused 4 casualties and an until then unseen damage of €12,9 billion (CRED/UCLouvain, 2023).

And this is the paragraph we will add to the main text after line 231 of the preprint:

> Very different levels of impacts were reported for these events. In section S2 of the supplementary material, we put each event in context to other available references (scientific or media), and also attempt to compile estimates of reported damages and loss of lives, if available. While all ten events featured exceptional amounts of rainfall and a corresponding runoff response, only five of them caused massive impacts (SN/Aug02, SN/Jun13, BW/May16, BB/Jun17, and, with by far the highest impact, NW/Jul21) while for the remaining events (LS/Jul02, LS/Jul17, LS/Jul17, HS/May19, and BB/Jun21), the impact was apparently not high enough to attract attention beyond the affected regions. The results of the counterfactual scenario analysis, as presented in the following, should help to understand whether the different levels of impacts for these events were mainly caused by their specific geographic position.

**RC:** *In some passages, I would wish more context for readers who are not familiar with the topic (or from other disciplines). (More complex) figures could be better introduced: what is the motivation of the figure or what exactly does the figure show before starting the interpretation (e.g. Fig 3, 4, 6). In addition, the authors do not take international readers into account – for them a figure with relevant names (cities, mountains) would be very helpful; this figure should also contain the orography that is relevant for the interpretation of Figure 5.*

AR: The suggestions are a very good basis to make the paper more comprehensible for various audiences. We will introduce corresponding changes to the main text and to the figure captions, and also introduce a new figure with a map that should introduce the study region. The changes are listed in the following:

- Change in the main text:

> Figure 3 illustrates the results from the counterfactual study for the Altenahr catchment. The total rainfall for the catchment for each counterfactual and the resulting highest QR peak is shown. Despite the positive correlation ($r = 0.96$, Fig. 3) between total rainfall and resulting flood peaks we notice that the same total rainfall amounts can yield markedly diverse QR peaks.

we added one sentence to the caption:

> Total rainfall amount and resulting QR peak for counterfactuals of the NW/Jul21 (yellow to blue) and BB/Jun2021 (grey) HPEs for the Altenahr catchment.

Figure 3: We added a little more information to the caption:

> Contributions of individual subbasins to the runoff peak at Altenahr for three scenarios. The left side shows the superposition of runoff from the subbasins. The color code describes the runoff contribution to the peak flow (white = low, red = high). On the right side, the same color code is used to display the spatial distribution of the contributions of each subbasin. Streams are shown in black as well as the outlet at Altenahr (black dot). Each row of the plot shows a different precipitation scenario a) original NW/Jul21 event, b) NW/Jul21_a counterfactual, c) BB/Jun21_a counterfactual (see also Tab. 2).

**RC:** *Please reflect your structure of the section 4. I had to read it twice to understand it. Please try to introduce everything before you start the interpretation (e.g. introduce complex figures before you discuss them). A clear sentence at the beginning (motivation for the following section, aim or scientific question to be investigated) could help the reader to understand more quickly what you are trying to achieve. You have nice results, but could discuss them in a more structured way.*

**AR:** We agree that section 4 should start with a gentler introduction that gives the reader an idea of the general line of thought. We will hence add the following paragraph after line 222 of the preprint.

> In this section, we present the results of our analysis. Section 4.1 starts by introducing the ten most severe precipitation events which were identified based on the cross-scale extremity index. By shifting them all over Germany, they form the basis of our spatial counterfactual search experiment. The hydrological simulation results of this experiment are first explored in a case study for the Ahr catchment, and put into context to the devastating flood event in July 2021 (section 4.2). Second, we summarize the results of our simulation experiment for all of Germany.

And while sections 4.2 and 4.3 already contain an introductory sentence to provide the context for the section, this could be improved for section 4.1 as follows:

> In this section, we introduce the ten most severe precipitation events between 2001 and 2022, based on DWD's RADKLIM dataset. These events are the basis of our counterfactual simulation experiment.

**RC:** *L321f: "The dataset can also be an helpful asset to identify flash flood hot spots in Germany." And what exactly would the hot spots be? Somehow the result is missing? Please work out better which regions would be particularly affected. At the moment you remain rather abstract. Maybe you should think about reducing the high resolution of Figure 5? Would a figure based on the "factors" be an idea (to show which postcode areas have been "too little" affected so far or where the danger could currently be underestimated too much due to the "missing" history)?*

**AR:** We agree that the sentence on "flash flood hot spots" is not clear enough. It should rather be as follows:

> Specifically, the results could be used as a basis to further explore the geographic variation of the flash flood hazard in more detail, and to identify sub-basins that appear particularly prone to flash floods, mainly as a result of topographical controls.

We would like to note that the above sentence does not refer to specific results discussed in this section, but should rather point out future use cases. That is why we would also prefer not to change the way the results are presented in Fig. 5 of the preprint, as this figure gives a strong impression of the spatial variability of the maximum UPD.

**RC:** *L68f: Suggestion: Here you can again clearly define or clarify the study area for the reader.*

AR: We have created a new subsection ("Study area") at the beginning of section 2 which introduces the study area, namely Germany (as the full study area) and the Ahr catchment as our case study area. For this new subsection, we have also adopted various other suggestions of the referee to which we will respond separately below.

**RC:** *Sect. 2.1. Prec data Which season? The whole year or just the summer half-year?*

AR: We used the entire dataset from 2001-2022. As it turned out, the top 10 events all occurred in the summer half-year. This makes perfect sense since events with high xWEI values, i.e. a high extremity across scales (including very high intensities at short durations), rather occur in the summer. Future applications could hence focus on the summer months in order to reduce the computational costs (which are substantial).

**RC:** *L105ff: What are the advantages/disadvantages of xWEI compared to CatRaRE; again emphasise in writing exactly what is considered in xWEI that is not considered in CatRaRE (soil moisture is indirectly implied but clearly communicated).*

AR: While we think that ll. 106-110 of the preprint already explained the reasons of why to prefer the xWEI over the WEI index in the context of this study, we agree that the reader is not sufficiently informed about how these indices actually differ from each other. We will add a corresponding explanation after line 112 of the preprint.

> Both the WEI (as used by the CatRaRE catalog) and the xWEI quantify a measure of extremeness along two dimensions: rainfall duration and spatial extent. Hence the variation of extremeness along these dimensions could be illustrated as a surface. While the WEI corresponds to the maximum value of that surface, the xWEI corresponds to the volume under the surface, meaning that it is high if the extremeness is high across spatial and temporal scales.

Further details with regard to both indices are comprehensively documented in the reference Voit and Heistermann (2022). Furthermore, we would like to emphasize that some level of arbitrariness cannot be avoided with regard to the choice of an index, even if we have good reasons to prefer the xWEI. In sections 5.1 and 6, we already emphasized that other events from our catalog could be selected as candidates for a counterfactual search. We will add one more sentence to section 5.1 (in l. 336) to clarify that also other indices or catalogs could be used for that purpose. Hence, the entire paragraph from ll. 334-337 will become:

> And, finally, the top ten events from our catalog might not yet represent the worst case in terms of the QR response at the "flash flood scale". Particularly for very small headwater catchments, other events from the catalog could trigger higher runoff peaks even if their xWEI were smaller. For prospective research, other severity indices, ranking criteria or catalogs might still be considered or developed which could provide a more explicit focus on flash floods and might hence serve to an even more exhaustive counterfactual search.

We like to underline once more that our study provides a *framework* for a downward counterfactual search for flash floods, but that each component (event selection, counterfactual scenario design, hydrological model) could be replaced by other approaches. This was already highlighted in section 6 of the preprint (final paragraph).

**RC:** *L117: "...high, if the rainfall in the 3 km x 3 km neighborhood was extreme at multiple durations (between 1 h and 72 h)." Perhaps add an example to make it easier for the reader to understand?*

AR: We have developed a figure in order to explain this better. As this figure, however, is quite bulky, and as not all readers might be interested in this level of technical detail, we suggest to shorten, in the main article, the explanation of how the catalog was generated. Instead, we provide a more detailed explanation, together with the new figure, in the supplementary material and refer to this in the main text.

Hence, we replaced ll. 113-126 of the preprint by the following shorter text:

> The catalog was created by applying a multi-step procedure. Considering the RADKLIM dataset as a 3-D array (one temporal dimension, two spatial dimensions), we first apply a moving 3-D window (72 hours x 3 km x 3 km) to the entire dataset. Within this moving window, the rainfall extremeness is computed for each voxel and for various durations. Afterwards, a clustering algorithm is applied to identify spatio-temporal clusters of extreme rainfall. The details of this approach together with an illustration are provided in the supplementary material.

In the supplementary, we suggest to use Fig. 2 for illustrating the process (you can find the figure at the end this PDF because it is quite large):

Referring to this figure, we will explain the process in the supplementary as follows:

The catalog was created as follows (see also Fig. S1) for illustration). For simplification we just used only two durations in Figure S1 (1 and 72 h), while in our actual study we used eight durations (1, 2, 4, 6, 12, 24, 48, 72 h):

1. We applied a 3 km x 3 km x 72 km) moving window for each pixel in the RADKLIM dataset. In Figure S1 a) and b) the pixel is surrounded by a red box. In this moving window we aggregate the rainfall to the durations to respective durations (Figure S1 c) and d)). For each duration we calculate the return periods for every pixel in the moving window (Figure S1 e) and f)). Now we can compute the xWEI. The return periods get sorted by decreasing order (Figure S1 g) and h)). We then compute the extremeness, $E_{tA}$ based on Müller et al., 2014:

$$E_{tA} = \frac{\sum_{i=1}^{n} ln(p_{t,i})}{n} * \frac{\sqrt{A}}{\sqrt{\pi}} \qquad [ln(year)km] \qquad (1)$$

The process is explained in more detail in Voit and Heistermann (2022).

Following this procedure, we get an $E_{tA}$-curve for every duration (Fig. S1 i) and j)). The $E_{tA}$-curves are placed on a grid (Fig. S1 k)). The $E_{tA}$-curves span a surface. The volume underneath that surface is the xWEI-value for the pixel (Fig. S1 l)) which is high, if the rainfall in the 3 x 3 km neighborhood was extreme at multiple durations (between 1 h and 72 h).

2. This way the xWEI-moving window works as a filter for the rainfall data. The result is a dataset of xWEI values with the same dimensions (x, y, time) as the RADKLIM dataset. An xWEI value of ten is approximately equal to an event that had a return period of around 10 years on one duration and at a spatial scale of 9 km².

3. All cells with an xWEI < 10 were discarded (set to NaN) to ensure that there are just cells remaining which signify extreme rainfall. The remaining adjacent cells were clustered based on their neighborhood (pixels within 10 km). This way we obtained distinct clusters where the rainfall must have been exceptionally high.

4. Finally, we determined the bounding box and computed the xWEI value for the entire bounding box, for each identified cluster.

**RC:** *L128: "The study area is Germany." Suggestion: Define this already at the beginning of section 2.*

AR: You are right, this statement will be moved to the beginning of section 2, and also extended further as follows:

In this section, we will describe the data that was used for the extraction of HPEs as well as the data sources for our hydrological model. The overall study area is Germany. We will also present a case study in which we focus on the catchment of the Ahr river down to the runoff gauge at Altenahr. In our hydrological model, this catchment consists of 37 subbasins (details of this case study are presented in section 4.2). Both the overall study area as well as the case study area are illustrated in Fig. 1.

**RC:** *L1334: "The median basin size is 11.55 km2." Idea/Suggestion: Add a histogram of all catchments; including quartile values (25 %/75% percentile).*

AR: Thank you for this suggestion. However, we think that providing the median and the interquartile range in the

main manuscript already provides a good idea of the sub-basin sizes. We will add Figure 3 with the histogram to the supplementary material. We also apologize that we were using a wrong value in the pre-print. The median basin size is not 11.55 km² but 12 km².

We are changing the sentence to:

> The median basin size is 12 km² (25th percentile: 6.9 km², 75th percentile: 20.2 km²). Figure S2 (supplementary) illustrates the distribution of subbasin sizes as a histogram.

**RC:** *L135: "case study 1" Do you have one case study or more? (because you write "1")*

 AR: Thank you for pointing out this mistake. We will remove the "1" and just refer to the "case study"

**RC:** *L167 "SCS-CN" method Reference? References for "established"*

 AR: We apologize, the references got lost with a new manuscript version. We will add the references in the revised manuscript as follows:

> We use the established SCS-CN (curve number) method (U.S. Department of Agriculture-Soil Conservation Service, 1972; Ponce and Hawkins, 1996; Natural Resources Conservation Service, 2004) to calculate the effective precipitation depending on soil, land use and antecedent wetness.

**RC:** *L206: Suggestion: Include the table in results section 4 (or 4.1) (together with the spatial results) and only refer to this section here.*

 AR: We agree that this table kind of anticipates the results. We will move the table to section 4.1, but already refer to it in section 3.3 in case readers already want to get an idea of these top 10 events.

**RC:** *L209: "within the HPE's bounding box (not just for the aligned subbasin)". Can this be better explained what is meant by this?*

 AR: We suggest to revise the sentence as follows:

> We then modelled the QR response for all sub-basins within the HPE's bounding box (not just for the subbasin to which we shifted the centroid of the HPE).

**RC:** *L212: "By following this procedure, we generated approximately 230,000 QR data sets across Germany of counterfactual runoff data sets." Can the calculation of the value 230,000 be better explained to the reader?*

 AR: We suggest the following revision of this sentence:

> By following this procedure, we generated approximately 230,000 counterfactual QR scenarios across Germany (23,000 sub-basins multiplied by 10 HPEs with their centroids shifted across all sub-basins).

**RC:** *Table 1: "The ID was constructed from an acronym that specifies the federal state" A non-German wouldn't understand this, let alone that the names of the cities mean anything to them; could the spatial allocation be better explained to the international reader?*

AR: We agree, that international readers might not be familiar with Germany in detail. Most likely, many Germans also won't know these, partially, small towns. For this reason we chose the federal state as identifier. The reader can now refer to a new Figure (Figure 5 in this response letter) to get an idea about the location and see the federal states, including their abbreviation.

We included the municipality where the highest rainfall was registered, in case the reader wants to investigate this event further.

RC: *Table1: Ranking (1, 2, ....) is not given; not all dates have been transformed into English.*

AR: The ranking is implicitly given, because the rows are sorted by the xWEI in decreasing order. To clarify this, we added a rank column. After looking at the table again, we cannot find any dates which are not in the right format. Could you specify this?

RC: *L216: "unit peak discharge (UPD)" - Reference?*

AR: The UPD is a widely used concept (e.g. Tarolli et al. (2013), Gaume et al. (2008), Amponsah et al. (2018)). We added a reference which summarizes the concept:

Castellarin, Attilio. "Probabilistic envelope curves for design flood estimation at ungauged sites." Water Resources Research 43.4 (2007).

RC: *L228f: "events with a large bounding box" Please specify what you mean by this.*

AR: The detailed description of the workflow of the creation of the event catalog which we provided now hopefully clarifies many questions. Nevertheless, we agree that the term "bounding box" in this context is unnecessarily confusing. We thereby suggest to change it to "events with a large spatial extent".

RC: *L229: "a large xWEI are likely to include smaller event clusters" Do you mean: "a large xWEI value are likely to include some smaller event clusters"?*

AR: Yes, we indeed mean the xWEI value. We will change this sentence to:

> However, events with a large spatial extent and a large xWEI value are likely to include smaller event clusters that are extreme at smaller spatio-temporal scales which exactly motivated the choice to rank events by the xWEI (see also Sect. 3.1).

RC: *L241/Figure 3: Please introduce the figure better in the main text (more explanation) before going into the result (as the figure is quite complex).*

AR: Thank you for this comment. We mentioned the change in a previous answer.

> Figure 3 illustrates the results from the counterfactual study for the Altenahr catchment. The total rainfall for the catchment for each counterfactual and the resulting highest QR peak is shown. Despite the positive correlation ($r = 0.96$, Fig. 4) between total rainfall and resulting flood peaks we notice that the same total rainfall amounts can yield markedly diverse QR peaks.

RC: *L247: "the QR peak can vary by a factor of 2." Can you give a reason (or hypothesis) why this is the case?*

AR: We hope to illustrate this in the new version of Figure 4 (which is discussed in the answer to the following

referee comment). The spatio-temporal distribution of rainfall has a large influence on the peak formation, leading to remarkably different peaks.

**RC:** *Figure 4: Can you explain one or two sentences about "stream order" for the reader who is not so familiar with hydrology?*

After your remark, we looked at the Figure again and revised it entirely. Our main intention with this Figure is to show the range of possible flood peaks due to the spatio-temporal signature of the rainfall and the possible superposition of peaks from different subbasins. We decided to not use stream orders anymore. The Figure now shows the contribution of the subbasins to the flood peak at Altenahr. We think that the new figure is now much clearer and more informative. In this regard, we also decided to just show 3 cases, and leave out NW/Jul21_b because it clearly has a different structure of precipitation and does not show episodes of high intensity rainfall. Besides, NW/Jul_b is not a "downward" counterfactual because of the rather low peak. This will simplify section 4.1.

**RC:** *L254: Location of NWJul21_b events? How far away?*

AR: As previously mentioned we decided to leave out NW/Jul_b from section 4.1 now. The NW/Jul21_a counterfactual resulted from a spatial shift of only 75 km². We will add this information to section 4.1.

**RC:** *258f: "In our model, the catchment consist of 37 subbasins (Fig. 2a)." This information would be helpful to the reader before.*

AR: Thank you, this information is now included in the beginning of section 2.

**RC:** *L259f: "Using the same approach, we are also able to take the other nine events from Table 1 and compare the resulting QR flood peaks" Please be more specific; you mean that you apply the other observed cases to the Ahr area to see that...*

AR: Thank your for pointing out that this sentence is not quite clear. We suggest to change it to:

> By spatially shifting the other nine HPEs from Table 1 across Germany, we can get an idea of the kind of QR flood peaks that these HPEs could have triggered at Altenahr - had they happened in the region.

**RC:** *L263f: "have apparently caused even higher QR peaks in Altenahr" Can you hypothesize why this is the case?*

AR: In our opinion, a detailed analysis of these Top10 events and why they triggered so diverse run off scenarios is beyond the scope of this study (see also the next referee comment).

**RC:** *L266f: "Among these, there are no counterfactuals of the events BW/May16, BB/Jun17, LS/Jul17, HS/May19 and BB/Jun20" Why?*

AR: We agree, that this is an important follow-up question. In our opinion, further research is needed to answer this. Please also see our response below with regard to the referee's comment on Table 3 LL314ff.

We hence changed the sentence to:

Among these, there are no counterfactuals of the events BW/May16, BB/Jun17, LS/Jul17, HS/May19 and BB/Jun20. Further investigation is needed to understand the differences in the spatio-temporal structure of these events and how these HPEs were different to the other top 10 events to understand why these HPEs did not have the potential to create any maximum counterfactual peaks.

**RC:** *L288f: "Mountain and low mountain ranges such as the Harz mountains, Ore mountains, the Black Forest" International readers need a figure that contains relevant location information; this could also include the orography so that the relationship is clear to them.*

AR: We agree, that the international reader might not be able to relate to these locations. Therefore, we suggest to drop the names of the areas entirely and just mention "mountain ranges". Following your suggestion, we will add Fig. 5 at the beginning of section 2.1 which clearly shows the topography of Germany, the names of the federal states, some major cities and rivers, as well as a zoom to the Ahr catchment upstream Altenahr. We hope that this figure will help the readers to contextualize the results of our study.

We modified the mentioned sentence to:

Mountain and low mountain ranges (compare to Fig. 1) display high QR peaks and therefore high UPD in the downward counterfactual analysis.

**RC:** *Figure 6: I don't understand the figure; please provide a more precise caption. A caption should first explain the figure (in a scientific paper). You start with results.*

AR: We hope that we could make the figure caption more comprehensible (as follows):

Cumulative distribution of the ratio between the highest counterfactual and the highest original peak for every subbasin is shown in red (cf_germany). From yellow to orange we show the same ratio but for counterfactuals with a limited shifting distance (10, 20, 50, 250 km, see section 3.3). QR peaks resulting from counterfactual simulations are much higher than the QR peaks caused by original events. As the shifting distance increases, more counterfactuals are considered for each subbasin. As a results, it becomes more likely that the counterfactual peaks are substantially higher than the highest original peak.

**RC:** *Table 3 & LL314ff: "Table 3 shows which events caused the highest discharges for sub-basins all across Germany: the counterfactuals of the event SN Jun/13 have caused the highest QR peaks in 82 % of the subbasins. Out of the ten HPEs, this is also the event with the highest hourly precipitation rates (see Tab. 1). Only in two subbasins, the highest QR peaks were caused by NW/Jul21 counterfactuals. In only one case, the worst case scenario was caused by an original event." And what does that mean; can you elaborate on the highlight of this statement? Can you be more specific? What do we learn from this finding?*

AR: We agree that it would be interesting to better understand which specific features of an event govern the runoff response and hence determine the ranking of events in Tab. 3. However, we are convinced that this requires a more detailed investigation that should not only include the analysis of specific constellations that lead to extreme runoff responses, but also a more comprehensive counterfactual search. Furthermore, such an investigation should not only focus on the differences between the events, but also on the similarities (both in rainfall structure and simulated runoff response). Such an investigation is beyond the scope of the present

exploratory study, and any attempt to explain the response to specific events will remain speculative until further analysis is presented.

In order to emphasize the above points, we have revised the corresponding paragraph (after l. 314 of the preprint) as follows:

> [...] Table 3 shows which events caused the highest discharges for sub-basins all across Germany: the counterfactuals of the event SNJun/13 have caused the highest QR peaks in 82 % of the subbasins. Out of the ten HPEs, this is also the event with the highest hourly precipitation rates (see Tab. 1). Then again, the BB/Jun21 event also accounts for a substantial proportion of maximum counterfactual peaks while it only ranks sixth with regard to hourly precipitation levels. Only in two subbasins, the highest QR peaks were caused by NW/Jul21 counterfactuals. In only one case, the worst case scenario was caused by an original event. While we expect the maximum counterfactual peaks to be governed by the interaction of specific spatio-temporal HPE features and basin properties, the nature of this interaction remains yet to be explained. In other words, it should be subject to future research to better understand which features favour an exceptional runoff response at the flash-flood scale. Such research should not be limited to the top 10 events, but aim for a more comprehensive counterfactual search (see section 5.1).

**RC:** *Section 5: A brief introduction and motivation for the section would be helpful to the reader. Perhaps rename section 5 to "Discussion" or "Discussion and Limitations"?*

 AR: We suggest "Uncertainties and limitations" as a new section title. In addition, we will add an introductory sentence after l. 323 (of the preprint) that should help the reader to better understand the motivation of this section.

> In this section, we highlight the uncertainties and limitations that should be kept in mind when interpreting the above results.

**RC:** *L342ff: This section seems misplaced here; suggest moving it to 4.1.*

 AR: We'd prefer to keep this paragraph in section 5.1 as it outlines an intrinsic limitation of the underlying dataset (in that it confines the representation of heavy precipitation events to the range of DWD's weather radar network). This leads to an underestimation of an event's actual extremeness in case it substantially extends over the German borders. In order to clarify the idea of this paragraph, we revised it as follows:

> Some HPEs, e.g. the SN/Aug2002 or the NW/Jul21 event are not completely captured by the DWD's weather radar network, as they extended across the borders of Germany. For these events, the extremeness is necessarily underestimated. We still decided to use these HPEs in our counterfactual simulation experiment because they are, even while being incompletely captured, among the 10 most extreme HPEs observed in Germany within the last 22 years.

**RC:** *L356: "Together with the expected underestimation of rainfall (see section 5.1)" What order of magnitude are we talking about here?*

We do not have comprehensive reports on the underestimation of the RADKLIM product. For DWD's operational radar-based QPE product (RADOLAN), there are some studies which report an underestimation

of up to 30% (Bronstert et al., 2018). We added this information to section 5.1; however, it should be emphasized that this kind of studies is only intermittent and not representative.

[revised manuscript text omitted]